# Diffusion Posterior Sampling for Linear Inverse Problem Solving — A Filtering Perspective

**Zehao Dou**
Yale University
`zehao.dou@yale.edu`

**Yang Song**
OpenAI
`songyang@openai.com`

## Abstract

Diffusion models have achieved tremendous success in generating high-dimensional data like images, videos and audio. These models provide powerful data priors that can solve linear inverse problems in zero shot through Bayesian posterior sampling. However, exact posterior sampling for diffusion models is intractable. Current solutions often hinge on approximations that are either computationally expensive or lack strong theoretical guarantees. In this work, we introduce an efficient diffusion sampling algorithm for linear inverse problems that is guaranteed to be asymptotically accurate. We reveal a link between Bayesian posterior sampling and Bayesian filtering in diffusion models, proving the former as a specific instance of the latter. Our method, termed *filtering posterior sampling*, leverages sequential Monte Carlo methods to solve the corresponding filtering problem. It seamlessly integrates with all Markovian diffusion samplers, requires no model re-training, and guarantees accurate samples from the Bayesian posterior as particle counts rise. Empirical tests demonstrate that our method generates better or comparable results than leading zero-shot diffusion posterior samplers on tasks like image inpainting, super-resolution, and motion deblur.

## 1 Introduction

Score-based diffusion models (Sohl-Dickstein et al., 2015; Song & Ermon, 2019; 2020; Ho et al., 2020; Song et al., 2020b) have made remarkable strides in data synthesis over the past few years, revolutionizing fields like image synthesis (Dhariwal & Nichol, 2021; Nichol et al., 2021; Ramesh et al., 2022; Saharia et al., 2022; Rombach et al., 2022; Zhang et al., 2023), video generation (Ho et al., 2022b;a), audio synthesis (Kong et al., 2020; Chen et al., 2020) and molecular conformation generation (Xu et al., 2022; Shi et al., 2021; Luo et al., 2021). As evidenced by their unparalleled sample quality, diffusion models provide powerful data priors that can capture the intricacies of high dimensional data distributions. Using such priors, we can deduce data from lossy measurements through Bayesian posterior sampling. This enables many diffusion-based methods for solving linear inverse problems without task-specific training (Song et al., 2020b; Chung et al., 2022c; Chung & Ye, 2022; Choi et al., 2021; Kawar et al., 2022; Chung et al., 2022a; Kawar et al., 2021; Bardsley et al., 2014; Song et al., 2023a; Jalal et al., 2021). Examples of their applications include image inpainting, colorization, super-resolution, motion deblur, and medical image reconstruction (Song et al., 2021).

Despite the empirical success of diffusion models in solving inverse problems, obtaining exact Bayesian posterior samples for these models remains intractable, necessitating the use of approximations. Current methods for this approximate sampling can be broadly grouped into three categories. The first approach modifies a standard diffusion sampling process by enforcing data consistency at every time step, ensuring all intermediate samples align with observed, lossy data measurements (Song et al., 2020b; Chung et al., 2022c; Chung & Ye, 2022; Choi et al., 2021; Kawar et al., 2022). The second approach estimates the score function (*i.e.*, the gradient of the log probability density) of the Bayesian posterior (Song et al., 2020b), leveraging it to guide each diffusion sampling iteration (Chung et al., 2022a; Kawar et al., 2021; Bardsley et al., 2014; Song et al., 2023a; Jalal et al., 2021; Song et al., 2023b). Lastly, some methods train a neural network to minimize a statistical divergence between its sample distribution and the true Bayesian posterior of the diffusion model (Graikos

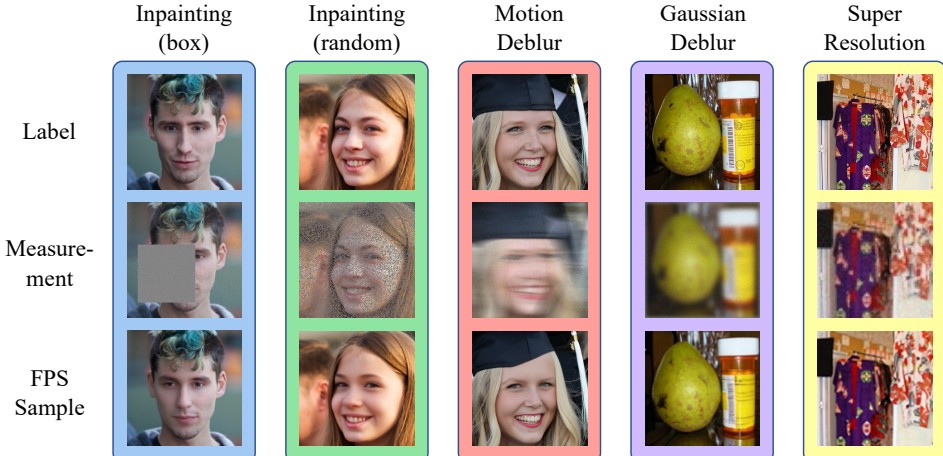

Figure 1: Several widely-used noisy measurements for images, and the recovered images by FPS.

et al., 2022; Feng et al., 2023; Feng & Bouman, 2023; Mardani et al., 2023). For the first two strategies, while each individual sampling iteration is grounded in theoretical insights, assessing the accumulated errors introduced over successive iterations poses a challenging theoretical question. The third strategy requires training additional neural networks for each novel task, which causes substantial computational overhead especially when applying the same diffusion prior to diverse inverse problems.

To address these challenges, we introduce a novel family of methods that allow diffusion models to tackle linear inverse problems in zero shot. We present a unique interpretation of Bayesian posterior sampling in diffusion models and establish its equivalence to Bayesian filtering. This insight lets us approach the Bayesian posterior sampling problem from a filtering perspective. Our proposed framework, *Filtering Posterior Sampling* (FPS), transforms any filtering algorithm into a diffusion posterior sampler. In particular, we devise an FPS algorithm based on particle filtering and sequential Monte Carlo (Doucet et al., 2001). To minimize the variance of particle filters, we tailor the proposal distribution at each sampling iteration. Our FPS algorithm sidesteps the need for costly optimization or training new neural networks. It works in harmony with any stochastic Markovian sampler of diffusion models and is easily adaptable to diverse linear inverse problems. As the particle count increases, our algorithm yields better posterior samples, offering a flexible trade-off between compute and sample quality.

Theoretically, we prove that our FPS algorithm correctly samples from the Bayesian posterior distribution as the number of particles approaches infinity when both the score estimator and SDE solver are perfect. To our knowledge, this is one of the first diffusion posterior sampling algorithms with a global consistence guarantee for general Bayesian linear inverse problems, along with concurrent works like Wu et al. (2023); Trippe et al. (2022). Furthermore, when using just one particle, we prove that our algorithm provides a specific approximation of the Bayesian posterior's score function at each sampling iteration, elucidating the link between FPS and existing methodologies. Empirically, we benchmark our FPS against current methods across various linear inverse problems in computer vision, such as inpainting (Yeh et al., 2017), super-resolution (Ledig et al., 2017; Haris et al., 2018) and motion deblur (Kupyn et al., 2019; Suin et al., 2020). In evaluations on both the FFHQ (Karras et al., 2019) and ImageNet (Deng et al., 2009) datasets, our FPS surpasses most competing methods.

## 2 BACKGROUND

**Linear Inverse Problems**  Linear inverse problems are pervasive in various scientific domains, with examples like image inpainting, colorization, super resolution, deblurring, Computed Tomography, and Magnetic Resonance Imaging. Given a datapoint $\mathbf{x}$, its lossy measurement is denoted as $\mathbf{y} = \boldsymbol{A}\mathbf{x} + \mathbf{n}$. In this equation, $\mathbf{x} \in \mathbb{R}^D$, $\mathbf{y} \in \mathbb{R}^d$, $\boldsymbol{A} \in \mathbb{R}^{d \times D}$, and $\mathbf{n} \sim \mathcal{N}(0, \sigma^2 \boldsymbol{I})$ represents the measurement noise with a known noise level $\sigma > 0$. The challenge of a linear inverse problem is to recover $\mathbf{x}$ from the incomplete measurement $\mathbf{y}$. As often $D > d$, this results in an ill-posed and under-

determined problem. To narrow down the solution space, we usually require additional information about $\mathbf{x}$. A common assumption is the prior knowledge of the distribution over $\mathbf{x}$, represented as $p(\mathbf{x})$. Given this, we can traverse the solution space of the linear inverse problem by sampling from the posterior distribution $p(\mathbf{x} \mid \mathbf{y})$, formulated using Bayes' rule as $p(\mathbf{x} \mid \mathbf{y}) \propto p(\mathbf{x})p(\mathbf{y} \mid \mathbf{x})$, with $p(\mathbf{y} \mid \mathbf{x}) = \mathcal{N}(\mathbf{y} \mid \boldsymbol{A}\mathbf{x}, \sigma^2\boldsymbol{I})$.

**Diffusion Models**   Diffusion models comprise a forward noising process and a backward denoising process. In the discrete formulation (Sohl-Dickstein et al., 2015; Ho et al., 2020), the forward process manifests as a Markov chain described by

$$q(\mathbf{x}_{1:N} \mid \mathbf{x}_0) = \prod_{k=1}^{N} q(\mathbf{x}_i \mid \mathbf{x}_{k-1}), \ \ \text{with } q(\mathbf{x}_k \mid \mathbf{x}_{k-1}) = \mathcal{N}\left(a_k\mathbf{x}_{k-1}, \ b_k^2\boldsymbol{I}\right). \tag{1}$$

The coefficients $\{a_k\}_{k=1}^{N}$ and $\{b_k\}_{k=1}^{N}$ are manually designed and may differ across various diffusion formulations (Song et al., 2020b). Given that each Markov step $q(\mathbf{x}_k \mid \mathbf{x}_{k-1})$ is a linear Gaussian model, the resultant marginal distribution $q(\mathbf{x}_k \mid \mathbf{x}_0) = \mathcal{N}(c_k\mathbf{x}_0, d_k^2\boldsymbol{I})$ assumes a Gaussian form, where $\{c_k\}_{k=1}^{N}$ and $\{d_k\}_{k=1}^{N}$ can be derived from $\{a_k\}_{k=1}^{N}$ and $\{b_k\}_{k=1}^{N}$. For sample generation, we train a neural network, denoted by $\boldsymbol{s}_{\boldsymbol{\theta}}(\mathbf{x}_k, t_k)$, to estimate the score function $\nabla_{\mathbf{x}_k} \log q(\mathbf{x}_k \mid \mathbf{x}_0)$. The backward process, which we assume to be a Markov chain, is typically represented as

$$p_{\boldsymbol{\theta}}(\mathbf{x}_{k-1} \mid \mathbf{x}_k) = \mathcal{N}\left(u_k\hat{\mathbf{x}}_0(\mathbf{x}_k) + v_k\boldsymbol{s}_{\boldsymbol{\theta}}(\mathbf{x}_k, t_k), w_k^2\boldsymbol{I}\right), \tag{2}$$

where $\hat{\mathbf{x}}_0(\mathbf{x}_k) := (\mathbf{x}_k + d_k^2\boldsymbol{s}_{\boldsymbol{\theta}}(\mathbf{x}_k, t_k))/c_k$ is the predicted $\mathbf{x}_0$ derived by $\mathbf{x}_k$ obtained from the Tweedie's formula. Here $\{u_k\}_{k=1}^{N}$, $\{v_k\}_{k=1}^{N}$ and $\{w_k\}_{k=1}^{N}$ can be computed from the forward process coefficients $\{a_k\}_{k=1}^{N}$ and $\{b_k\}_{k=1}^{N}$ in Eq. (1). $\{t_k\}_{k=0}^{N}$ is the set of discrete timestamps where $t_0 = 0, t_N = T$. The formulation in Eq. (2) encompasses many stochastic samplers of diffusion models, including the ancestral sampler in DDPM (Sohl-Dickstein et al., 2015; Ho et al., 2020), the predictor-corrector sampler in Song et al. (2020b), and the DDIM sampler in (Song et al., 2020a).

For variance preserving diffusion models (Ho et al., 2020), we have

$$a_k = \sqrt{\alpha_k}, \ b_k = \sqrt{\beta_k}, \ c_k = \sqrt{\overline{\alpha}_k}, \ d_k = \sqrt{1 - \overline{\alpha}_k},$$

where $\alpha_k := 1 - \beta_k, \overline{\alpha}_k := \prod_{j=1}^{k} \alpha_j$, and $\alpha_k, \beta_k$ follow the notations in Ho et al. (2020). To represent DDPM sampling, we have:

$$u_k = \sqrt{\alpha_{k-1}}, \ v_k = -\sqrt{\alpha_k}(1 - \overline{\alpha}_{k-1}), \ w_k = \sqrt{\beta_k \cdot \frac{1 - \overline{\alpha}_{k-1}}{1 - \overline{\alpha}_k}},$$

and for DDIM sampling (Song et al., 2020a), we have:

$$u_k = \sqrt{\alpha_{k-1}}, \ v_k = -\sqrt{1 - \overline{\alpha}_{k-1} - \sigma_k^2} \cdot \sqrt{1 - \overline{\alpha}_k}, \ w_k = \sigma_k$$

where the conditional variance sequence $\{\sigma_k\}_{k=1}^{N}$ can be arbitrary. Note that DDPM sampling is a special case of DDIM when setting $\sigma_k^2 = \beta_k \cdot (1 - \overline{\alpha}_{k-1})/(1 - \overline{\alpha}_k)$. In this paper, we parameterize

$$\sigma_k = c \cdot \sqrt{\beta_k \cdot \frac{1 - \overline{\alpha}_{k-1}}{1 - \overline{\alpha}_k}}$$

with $c$ being a tunable hyper-parameter.

## 3   BAYESIAN POSTERIOR SAMPLING AS BAYESIAN FILTERING

In the following part, we introduce Bayesian filtering and explain how it is related to Bayesian posterior sampling. Bayesian filtering aims to infer latent variables from observations based on a sequence of conditional probability distributions. For each time step $k = 1, 2, \ldots, N$, we define the latent state and its corresponding measurement as:

$$\mathbf{x}_k \sim p(\mathbf{x}_k \mid \mathbf{x}_{k-1}), \ \ \mathbf{y}_k \sim p(\mathbf{y}_k \mid \mathbf{x}_k).$$

The dynamic model of this system determines $\mathbf{x}_k$ and $\mathbf{y}_k$ depends on the distribution of measurements given the state. As we can see, the sequence $\{\mathbf{x}_k\}_{k=0}^{N}$ forms a Markov chain. The Bayesian filtering problem seeks to sample from the distribution $p(\mathbf{x}_k \mid \mathbf{y}_{1:k})$.

To derive the conditional distribution, a well-known iterative approach is utilized. Starting with the known prior distribution $p(\mathbf{x}_0)$, we employ the Chapman-Kolmogorov equation to obtain $p(\mathbf{x}_k \mid \mathbf{y}_{1:k})$ from $p(\mathbf{x}_{k-1} \mid \mathbf{y}_{1:k-1})$ as the "prediction" step

$$p(\mathbf{x}_k \mid \mathbf{y}_{1:k-1}) = \int p(\mathbf{x}_k \mid \mathbf{x}_{k-1}) \cdot p(\mathbf{x}_{k-1} \mid \mathbf{y}_{1:k-1}) d\mathbf{x}_{k-1}, \tag{3}$$

and then use Bayes' rule to compute $p(\mathbf{x}_k \mid \mathbf{y}_{1:k})$ as the "update" step:

$$p(\mathbf{x}_k \mid \mathbf{y}_{1:k}) = \frac{p(\mathbf{y}_k \mid \mathbf{x}_k) \cdot p(\mathbf{x}_k \mid \mathbf{y}_{1:k-1})}{\int p(\mathbf{y}_k \mid \mathbf{x}_k) \cdot p(\mathbf{x}_k \mid \mathbf{y}_{1:k-1}) d\mathbf{x}_k} \tag{4}$$

By iterating these two steps, we can solve the Bayesian filtering problem. For those system where dynamics and measurements adhere to linear Gaussian distributions, Kalman filter (Kalman, 1960) provides a closed form solution to the Bayesian filtering problem.

Back to our diffusion posterior sampling problem, the dynamic system's backward process is given by $\mathbf{x}_k \sim p_{\boldsymbol{\theta}}(\mathbf{x}_k \mid \mathbf{x}_{k+1})$, where $p_{\boldsymbol{\theta}}(\mathbf{x}_k \mid \mathbf{x}_{k+1})$ is defined in Eq. (2). The forward process is described by

$$\mathbf{x}_k = a_k \cdot \mathbf{x}_{k-1} + b_k \mathbf{z}_k \tag{5}$$

where $\mathbf{z}_k \sim \mathcal{N}(0, \boldsymbol{I})$ is a standard Gaussian noise independent of $\mathbf{x}_{k-1}$. Meanwhile, we construct the $\{\mathbf{y}_k\}_{k=0}^N$ sequence as:

$$\mathbf{y}_0 = \mathbf{y}, \quad \mathbf{y}_k = a_k \cdot \mathbf{y}_{k-1} + b_k \cdot \boldsymbol{A}\mathbf{z}_k \text{ for } k \geqslant 1. \tag{6}$$

The noise sharing technique is also applied in Song et al. (2021), but with a different framework. Given $\mathbf{y} \sim \mathcal{N}(\boldsymbol{A}\mathbf{x}_0, \sigma^2\boldsymbol{I})$, it is easy to show that $\mathbf{y}_k \sim \mathcal{N}(\boldsymbol{A}\mathbf{x}_k, c_k^2\sigma^2\boldsymbol{I})$ for $c_k = a_1 a_2 \ldots a_k$ through mathematical induction. This formulation establishes Bayesian posterior sampling as a reverse-time Bayesian filtering problem, a well-studied topic with many existing solutions and algorithms. By following the prediction and update steps in Bayesian filtering, we successively compute the marginal posterior distribution $p_{\boldsymbol{\theta}}(\mathbf{x}_k \mid \mathbf{y}_{k:N})$ until we obtain $p_{\boldsymbol{\theta}}(\mathbf{x}_0 \mid \mathbf{y}_{0:N})$. This allows us to sample from the Bayesian posterior $p_{\boldsymbol{\theta}}(\mathbf{x}_0 \mid \mathbf{y}_0)$ due to the following important observation:

> We can generate a sample $\mathbf{x}_0$ from the Bayesian posterior distribution $p_{\boldsymbol{\theta}}(\mathbf{x}_0 \mid \mathbf{y}_0)$ by first sampling from $p_{\boldsymbol{\theta}}(\mathbf{y}_{1:N} \mid \mathbf{y}_0) = q(\mathbf{y}_{1:N} \mid \mathbf{y}_0)$ before sampling from $p_{\boldsymbol{\theta}}(\mathbf{x}_0 \mid \mathbf{y}_{0:N})$. This is because $p_{\boldsymbol{\theta}}(\mathbf{x}_0 \mid \mathbf{y}_0) = \int p_{\boldsymbol{\theta}}(\mathbf{x}_0 \mid \mathbf{y}_{0:N}) \cdot p_{\boldsymbol{\theta}}(\mathbf{y}_{1:N} \mid \mathbf{y}_0) d\mathbf{y}_{1:N}$.

Here, $q$ and $p_{\boldsymbol{\theta}}$ denote the probability distribution of the forward process and the backward process respectively. That is, we can leverage any algorithm for Bayesian filtering to solve the Bayesian posterior sampling problem whenever the prior is captured by a diffusion model.

## 4 FILTERING POSTERIOR SAMPLING (FPS)

In this section, we propose a specific algorithm for solving the Bayesian filtering problem associated with a diffusion model. Due to the aforementioned connection between Bayesian filtering and Bayesian posterior sampling, this algorithm can be directly employed to generate posterior samples from a diffusion model for inverse problem solving. Our algorithm is based on particle filtering, but has a tailored proposal distribution for variance reduction.

### 4.1 BACKWARD DIFFUSION PROCESS WITH LINEAR FILTERING

In the backward diffusion process, we first generate the sequence of $\{\mathbf{y}_k\}_{k=0}^N$. Then, we recursively sample $\mathbf{x}_{k-1}$ based on $\mathbf{x}_k$ and $\mathbf{y}_{k-1}$ by solving Bayesian linear filtering.

**Step 1: Generating Sequence** $\{\mathbf{y}_k\}_{k=0}^N$     There are two different ways to generate the sequence of $\{\mathbf{y}_k\}_{k=0}^N$. One leverages the forward process and the other depends on the backward process. For the forward process, we apply Eq. (6) to obtain

$$\mathbf{y}_k = a_k \mathbf{y}_{k-1} + b_k \cdot \boldsymbol{A}\mathbf{z}_k$$

where $\mathbf{z}_k \sim \mathcal{N}(0, \boldsymbol{I})$ is independent Gaussian noise, and the initial observation $\mathbf{y}_0 = \mathbf{y}$ is given. When generating $\{\mathbf{y}_k\}_{k=0}^N$ from the backward process, we apply DDIM sampling. Since we have full access to $\mathbf{y}_0$, there is no need to predict $\hat{\mathbf{y}}_0(\mathbf{y}_k)$ because $\hat{\mathbf{y}}_0(\mathbf{y}_k) = \mathbf{y}_0$. Therefore, we can sample $\{\mathbf{y}_k\}_{k=0}^N$ via the following recurrence relation:

$$\mathbf{y}_{k-1} = u_k \mathbf{y}_0 + v_k \mathbf{y}_k + w_k \cdot \boldsymbol{A}\mathbf{z}_k. \tag{7}$$

where $u_k, v_k, w_k$ can be computed from $a_k, b_k$ (see Section 2), and the initial observation $\mathbf{y}_N$ approximately follows $\mathcal{N}(\mathbf{0}, \boldsymbol{A}\boldsymbol{A}^\top)$. As a result, the sequence $\{\mathbf{y}_k\}_{k=0}^N$ is fully determined through either direction, given by

$$q(\mathbf{y}_{1:N} \mid \mathbf{y}_0) = \prod_{k=1}^N q(\mathbf{y}_k \mid \mathbf{y}_{k-1}, \mathbf{y}_0) = q(\mathbf{y}_N) \cdot \prod_{k=2}^N q(\mathbf{y}_{k-1} \mid \mathbf{y}_k, \mathbf{y}_0). \tag{8}$$

**Remark 4.1.** *The generation of sequence $\{\mathbf{y}_k\}_{k=0}^N$ from $\mathbf{y}_0$ is possible only in linear inverse problems. Otherwise, for nonlinear inverse problems, Eq. (6) not longer works, and neither forward or backward processes of $\{\mathbf{y}_k\}_{k=0}^N$ is tractable.*

**Step 2: Backward Sequence of $\{\mathbf{x}_k\}_{k=0}^N$** In the second step, we generate the backward sequence of $\{\mathbf{x}_k\}_{k=0}^N$ based on the measurement sequence $\{\mathbf{y}_k\}_{k=0}^N$ generated in Step 1, which leads to $p_{\boldsymbol{\theta}}(\mathbf{y}_{1:N} \mid \mathbf{y}_0) = q(\mathbf{y}_{1:N} \mid \mathbf{y}_0)$. Given that the last state $\mathbf{x}_N$ is approximately a standard Gaussian, we have

$$\mathbf{x}_N \sim p_{\boldsymbol{\theta}}(\mathbf{x}_N \mid \mathbf{y}_N) \propto p_{\boldsymbol{\theta}}(\mathbf{x}_N) \cdot p_{\boldsymbol{\theta}}(\mathbf{y}_N \mid \mathbf{x}_N) \text{ with } p_{\boldsymbol{\theta}}(\mathbf{x}_N) = \mathcal{N}(\mathbf{0}, \boldsymbol{I}).$$

Since $p_{\boldsymbol{\theta}}(\mathbf{y}_N \mid \mathbf{x}_N) = q(\mathbf{y}_N \mid \mathbf{x}_N)\mathcal{N}(\boldsymbol{A}\mathbf{x}_N, c_N^2 \sigma^2 \cdot \boldsymbol{I})$ where $c_N = a_1 a_2 \ldots a_N$, the posterior distribution $p_{\boldsymbol{\theta}}(\mathbf{x}_N \mid \mathbf{y}_N)$ is Gaussian and can be expressed in closed form.

Next, we progressively sample $\mathbf{x}_{k-1}$ conditioned on $\mathbf{x}_k$ and $\mathbf{y}_{k-1}$. We know that $p_{\boldsymbol{\theta}}(\mathbf{y}_{k-1} \mid \mathbf{x}_{k-1}) = q(\mathbf{y}_{k-1} \mid \mathbf{x}_{k-1}) = \mathcal{N}(\boldsymbol{A}\mathbf{x}_{k-1}, c_{k-1}^2 \sigma^2 \cdot \boldsymbol{I})$ with $c_{k-1} = a_1 a_2 \ldots a_{k-1}$. In addition, $p_{\boldsymbol{\theta}}(\mathbf{x}_{k-1} \mid \mathbf{x}_k)$ is determined once we have the score function:

$$p_{\boldsymbol{\theta}}(\mathbf{x}_{k-1} \mid \mathbf{x}_k) = \mathcal{N}(u_k \hat{\mathbf{x}}_0(\mathbf{x}_k) + v_k \boldsymbol{s}_{\boldsymbol{\theta}}(\mathbf{x}_k, t_k), w_k^2 \boldsymbol{I})$$

where $\hat{\mathbf{x}}_0(\mathbf{x}_k) := (\mathbf{x}_k + d_k^2 \boldsymbol{s}_{\boldsymbol{\theta}}(\mathbf{x}_k, t_k))/c_k$ is the conditional expectation of $\mathbf{x}_0$ given $\mathbf{x}_k$, which can be computed via Tweedie's formula. Here, the choices of parameters $c_k, d_k, u_k, v_k, w_k$ are determined from the process of unconditional diffusion sampling. For the DDPM and DDIM framework, we list these parameters in Section 2. Now, we can compute the posterior distribution

$$p_{\boldsymbol{\theta}}(\mathbf{x}_{k-1} \mid \mathbf{x}_k, \mathbf{y}_{k-1}) \propto p_{\boldsymbol{\theta}}(\mathbf{x}_{k-1} \mid \mathbf{x}_k) \cdot p_{\boldsymbol{\theta}}(\mathbf{y}_{k-1} \mid \mathbf{x}_{k-1}). \tag{9}$$

Here, we rely on the conditional independence of $\mathbf{x}_k$ and $\mathbf{y}_{k-1}$ given $\mathbf{x}_{k-1}$. Since both $p_{\boldsymbol{\theta}}(\mathbf{x}_{k-1} \mid \mathbf{x}_k)$ and $p_{\boldsymbol{\theta}}(\mathbf{y}_{k-1} \mid \mathbf{x}_{k-1})$ are Gaussian, it is straightforward to conclude that $p_{\boldsymbol{\theta}}(\mathbf{x}_{k-1} \mid \mathbf{x}_k, \mathbf{y}_{k-1})$ is also a Gaussian distribution with a tractable form. Given $\{\mathbf{y}_k\}_{k=0}^N$, we can now sample from $p_{\boldsymbol{\theta}}(\mathbf{x}_{k-1} \mid \mathbf{x}_k, \mathbf{y}_{k-1})$ recursively for $k = N, N-1, \cdots, 1$ to obtain $\{\mathbf{x}_k\}_{k=0}^N$. This procedure provides an approximate solution to the Bayesian filtering problem, and we call this algorithm *Filtering Posterior Sampling* (FPS).

### 4.2 FURTHER ANALYSIS ON FPS

In this section, we analyze how FPS provides an approximate solution to the Bayesian filtering problem. As discussed in Section 3, we recursively compute a prediction step and an update step to calculate $p(\mathbf{x}_k \mid \mathbf{y}_{k:N})$. For FPS, the following approximation is made:

$$p_{\boldsymbol{\theta}}(\mathbf{x}_k, \mathbf{x}_{k+1} \mid \mathbf{y}_{k:N}) = p_{\boldsymbol{\theta}}(\mathbf{x}_{k+1} \mid \mathbf{y}_{k:N}) p_{\boldsymbol{\theta}}(\mathbf{x}_k \mid \mathbf{x}_{k+1}, \mathbf{y}_{k:N}) = p_{\boldsymbol{\theta}}(\mathbf{x}_{k+1} \mid \mathbf{y}_{k:N}) p_{\boldsymbol{\theta}}(\mathbf{x}_k \mid \mathbf{x}_{k+1}, \mathbf{y}_k)$$
$$\approx p_{\boldsymbol{\theta}}(\mathbf{x}_{k+1} \mid \mathbf{y}_{k+1:N}) \cdot p_{\boldsymbol{\theta}}(\mathbf{x}_k \mid \mathbf{x}_{k+1}, \mathbf{y}_k). \tag{10}$$

Here, we leverage the conditional independence between $\mathbf{y}_{k+1}$ and $\mathbf{y}_{k+2:N}$ when $\mathbf{x}_{k+1}$ is given. After taking integral on both sides of Eq. (10) with regard to $\mathbf{x}_{k+1}$, we have

$$p_{\boldsymbol{\theta}}(\mathbf{x}_k \mid \mathbf{y}_{k:N}) \approx \int p_{\boldsymbol{\theta}}(\mathbf{x}_{k+1} \mid \mathbf{y}_{k+1:N}) \cdot p_{\boldsymbol{\theta}}(\mathbf{x}_k \mid \mathbf{x}_{k+1}, \mathbf{y}_k) \mathrm{d}\mathbf{x}_{k+1}, \tag{11}$$

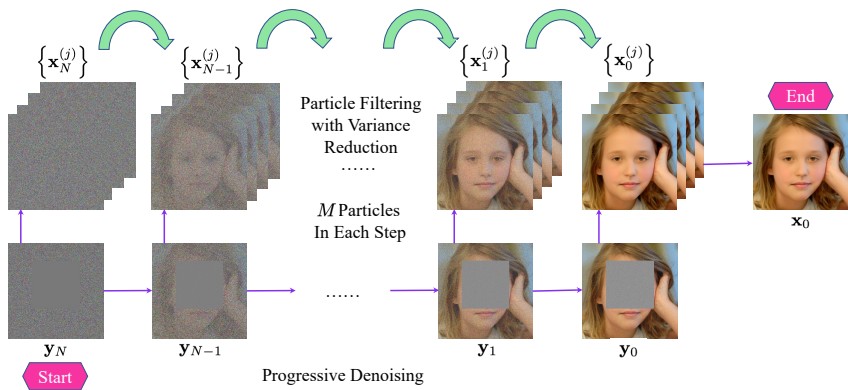

Figure 2: Illustration of FPS-SMC algorithm.

which means we skip the update step in FPS, and we make the approximation $p_{\boldsymbol{\theta}}(\mathbf{x}_{k+1} \mid \mathbf{y}_{k:N}) \approx p_{\boldsymbol{\theta}}(\mathbf{x}_{k+1} \mid \mathbf{y}_{k+1:N})$. Here, we sample a single particle $\mathbf{x}_{k+1}$ from $p_{\boldsymbol{\theta}}(\mathbf{x}_{k+1} \mid \mathbf{y}_{k+1:N})$ and then another particle $\mathbf{x}_k \sim p_{\boldsymbol{\theta}}(\mathbf{x}_k \mid \mathbf{y}_{k:N})$ by following the prediction step. Notice that Kalman filtering is not applicable here because $p_{\boldsymbol{\theta}}(\mathbf{x}_k \mid \mathbf{x}_{k+1})$ is not a linear Gaussian process, so Monte Carlo methods are necessary for solving this Bayesian filtering problem.

One can derive additional theoretical insights by generalizing FPS to continuous-time Markov chains. When we make the time step $\Delta t \to 0$ and consider the continuous limit of FPS, our backward diffusion process converges to the following SDE:

$$\mathrm{d}\mathbf{x}_t = \left[ -\frac{\beta(t)}{2}\mathbf{x}_t - \beta(t)\nabla_{\mathbf{x}_t}\log p_{\boldsymbol{\theta},t}(\mathbf{x}_t \mid \mathbf{y}_t) \right]\mathrm{d}t + \sqrt{\beta(t)}\mathrm{d}\overline{\mathbf{W}}_t.$$

For the detailed proof, see Appendix D.1. According to this equation, FPS approximates the unknown conditional score function $\nabla_{\mathbf{x}_t}\log p_{\boldsymbol{\theta},t}(\mathbf{x}_t \mid \mathbf{y})$ with $\nabla_{\mathbf{x}_t}\log p_{\boldsymbol{\theta},t}(\mathbf{x}_t \mid \mathbf{y}_t)$. Unlike the original score-based SDE, $\mathbf{y}_t$ is not sampled from a noisy version of $\mathbf{y}$ in FPS; instead, they are sampled from a backward diffusion process.

### 4.3 FPS WITH SEQUENTIAL MONTE CARLO

To improve the approximation made by FPS, we propose a variant based on particle filtering and sequential Monte Carlo. We call this improved method Filtering Posterior Sampling with Sequential Monte Carlo (FPS-SMC).

To begin with, notice that

$$\begin{aligned}
p_{\boldsymbol{\theta}}(\mathbf{x}_{k:N} \mid \mathbf{y}_{k:N}) &\propto p_{\boldsymbol{\theta}}(\mathbf{y}_k \mid \mathbf{x}_{k:N}, \mathbf{y}_{k+1:N}) \cdot p_{\boldsymbol{\theta}}(\mathbf{x}_{k:N} \mid \mathbf{y}_{k+1:N}) \\
&= p_{\boldsymbol{\theta}}(\mathbf{y}_k \mid \mathbf{x}_k)p(\mathbf{x}_k \mid \mathbf{x}_{k+1:N}, \mathbf{y}_{k+1:N}) \cdot p_{\boldsymbol{\theta}}(\mathbf{x}_{k+1:N} \mid \mathbf{y}_{k+1:N}) \\
&= p_{\boldsymbol{\theta}}(\mathbf{y}_k \mid \mathbf{x}_k)p_{\boldsymbol{\theta}}(\mathbf{x}_k \mid \mathbf{x}_{k+1}) \cdot p_{\boldsymbol{\theta}}(\mathbf{x}_{k+1:N} \mid \mathbf{y}_{k+1:N}) \\
&= \frac{p_{\boldsymbol{\theta}}(\mathbf{y}_k \mid \mathbf{x}_k)p_{\boldsymbol{\theta}}(\mathbf{x}_k \mid \mathbf{x}_{k+1})}{p_{\boldsymbol{\theta}}(\mathbf{x}_k \mid \mathbf{x}_{k+1}, \mathbf{y}_k)} \cdot p_{\boldsymbol{\theta}}(\mathbf{x}_k \mid \mathbf{x}_{k+1}, \mathbf{y}_k)p_{\boldsymbol{\theta}}(\mathbf{x}_{k+1:N} \mid \mathbf{y}_{k+1:N}).
\end{aligned} \tag{12}$$

In particle filtering, we generate $M$ i.i.d particles $\mathbf{x}_N^{(j)} \sim p_{\boldsymbol{\theta}}(\mathbf{x}_N \mid \mathbf{y}_N)$. After obtaining $\{\mathbf{x}_{k+1}^{(j)}\}_{j\in[M]}$, we move them backward for one step to obtain $\overline{\mathbf{x}}_k^{(j)} \sim p_{\boldsymbol{\theta}}(\mathbf{x}_k \mid \mathbf{x}_{k+1}^{(j)}, \mathbf{y}_k)$ for $j \in [M]$. Afterwards, we randomly sample $M$ particles with replacement, which we denoted as $\{\mathbf{x}_k^{(j)}\}_{j\in[M]}$. The particles are re-sampled according to the following probability distribution

$$\mathbb{P}\left[\mathbf{x}_k = \overline{\mathbf{x}}_k^{(j)}\right] := \eta_j = \frac{p_{\boldsymbol{\theta}}(\mathbf{y}_k \mid \overline{\mathbf{x}}_k^{(j)})p_{\boldsymbol{\theta}}(\overline{\mathbf{x}}_k^{(j)} \mid \mathbf{x}_{k+1}^{(j)})/p_{\boldsymbol{\theta}}(\overline{\mathbf{x}}_k^{(j)} \mid \mathbf{x}_{k+1}^{(j)}, \mathbf{y}_k)}{\sum_{j=1}^{M} p_{\boldsymbol{\theta}}(\mathbf{y}_k \mid \overline{\mathbf{x}}_k^{(j)})p_{\boldsymbol{\theta}}(\overline{\mathbf{x}}_k^{(j)} \mid \mathbf{x}_{k+1}^{(j)})/p_{\boldsymbol{\theta}}(\overline{\mathbf{x}}_k^{(j)} \mid \mathbf{x}_{k+1}^{(j)}, \mathbf{y}_k)}. \tag{13}$$

One important observation about the resampling weights is

$$\frac{p_{\boldsymbol{\theta}}(\mathbf{y}_k \mid \mathbf{x}_k)p_{\boldsymbol{\theta}}(\mathbf{x}_k \mid \mathbf{x}_{k+1})}{p_{\boldsymbol{\theta}}(\mathbf{x}_k \mid \mathbf{x}_{k+1}, \mathbf{y}_k)} = \frac{p_{\boldsymbol{\theta}}(\mathbf{y}_k \mid \mathbf{x}_k)p_{\boldsymbol{\theta}}(\mathbf{x}_k \mid \mathbf{x}_{k+1})}{p_{\boldsymbol{\theta}}(\mathbf{y}_k \mid \mathbf{x}_k, \mathbf{x}_{k+1})} = p_{\boldsymbol{\theta}}(\mathbf{y}_k \mid \mathbf{x}_{k+1}).$$

Based on this, we propose the FPS-SMC method with pseudo-code provided in Algorithm 1 and a visual illustration given in Fig. 2. Clearly, FPS can be viewed as a special case of FPS-SMC with particle size $M = 1$. Unlike FPS, FPS-SMC is consistent, meaning that the approximation error converges to zero as the particle size $M$ approaches infinity. Below we provide a rigorous statement about the convergence of FPS-SMC.

**Assumption 4.1.** *Assume the following two facts:*

(1) *Our score estimation is perfect, i.e: $s_\theta(\mathbf{x}_t, t) = \nabla_{\mathbf{x}_t} \log p_{\boldsymbol{\theta},t}(\mathbf{x}_t)$.*

(2) *The time step $\frac{T}{N} \to 0$, which means the backward diffusion ODE/SDE solver can provide accurate solution without discretization error.*

*These two facts make sure that both our diffusion model and sampling procedure are perfect during the backward Bayesian filtering.*

**Proposition 4.1.** *Denote $p_{\boldsymbol{\theta}}(\mathbf{x}_{0:N} \mid \mathbf{y}_{0:N})$ as the result of FPS-SMC algorithm with particle size $M$, and $p^*(\mathbf{x}_{0:N} \mid \mathbf{y}_{0:N})$ as the solution of Bayesian filtering problem (which follows the forward diffusion model). Under Assumption 4.1, we have $p_{\boldsymbol{\theta}}(\mathbf{x}_k \mid \mathbf{x}_{k+1}) = p^*(\mathbf{x}_k \mid \mathbf{x}_{k+1})$ for $\forall k = 0, 1, \ldots, N-1$. Then, we can conclude that:*

$$p_{\boldsymbol{\theta}}(\mathbf{x}_0 \mid \mathbf{y}_0) \overset{w}{\to} p^*(\mathbf{x}_0 \mid \mathbf{y}_0) = q(\mathbf{x}_0 \mid \mathbf{y}_0) \quad \text{when } M \to \infty,$$

*where $\overset{w}{\to}$ implies weak convergence.*

*Proof.* See Appendix D.2. $\square$

This result proves the asymptotic consistency of FPS-SMC when $M \to \infty, N \to \infty$. We can also derive finite sample bounds for $W_1\left(p_{\boldsymbol{\theta}}(\mathbf{x}_0 \mid \mathbf{y}_0), p^*(\mathbf{x}_0 \mid \mathbf{y}_0)\right)$ with regard to the estimation error in score models as well as the time step $\Delta t = T/N$. A more comprehensive theoretical exploration of these finite sample bounds remains a topic for our future research.

## 5 Experiments

### 5.1 Experimental Settings

We evaluate our algorithms on the FFHQ-1k-validation set (Karras et al., 2019) and the ImageNet-1k-validation set (Deng et al., 2009), both frequently used in inverse problem research involving diffusion models. Each dataset contains images of dimensions $256 \times 256 \times 3$, normalized to the range $[0, 1]$. These images serve for testing various linear inverse problems. We follow the experimental settings in Chung et al. (2022a) to ensure a fair comparison. We assume all the observations contain Gaussian noise of mean zero and standard deviation $\sigma = 0.05$. In our experiments for the FPS-SMC, we choose a particle size of $M = 20$. Another hyperparameter we find important for FPS and FPS-SMC is $c$, utilized during DDIM sampling. Our empirical observations highlight the significant influence of $c$ on the smoothness of generated images, so we tune $c$ separately for different tasks and datasets. For a more detailed analysis, see Appendix F for an ablation study, and Appendix E for a thorough description of all the five tasks we test in our experiments.

---

**Algorithm 1** The Framework of FPS-SMC

1: Given: $N, \mathbf{y}$, particle size $M$ and the score estimator $\boldsymbol{s}_{\boldsymbol{\theta}}(\cdot, \cdot)$.
2: Sample the $\{\mathbf{y}_k\}$ sequence from $q(\mathbf{y}_{1:N} \mid \mathbf{y}_0)$.

3: Sample $M$ i.i.d samples $\mathbf{x}_N^{(j)} \sim p_{\boldsymbol{\theta}}(\mathbf{x}_N \mid \mathbf{y}_N)$ for $j \in [M]$.
4: **for** $k = N, N-1, \ldots, 1$ **do**
5:     Generate $M$ i.i.d samples $\overline{\mathbf{x}}_{k-1}^{(j)} \sim p_{\boldsymbol{\theta}}(\mathbf{x}_{k-1} \mid \mathbf{x}_k^{(j)}, \mathbf{y}_{k-1})$ for $j \in [M]$.
6:     Randomly pick $M$ samples with replacement from $\left\{\overline{\mathbf{x}}_{k-1}^{(j)}\right\}_{j \in [M]}$, following the probability distribution (13). Denote the $M$ new samples as $\{\mathbf{x}_{k-1}^{(j)}\}_{j \in [M]}$.
7: **end for**
8: The output $\mathbf{x}_0$ is uniformly sampled from $\{\mathbf{x}_0^{(j)}\}_{j \in [M]}$.

---

During the backward process of FPS, we set the number of time steps as $N = 1000$ and use the pretrained score model from Chung et al. (2022a) for the FFHQ dataset, and the score model from Dhariwal & Nichol (2021) for the ImageNet dataset. We use multiple metrics to evaluate the quality of our generated images, including peak signal-to-noise ratio (PSNR), Fréchet Inception Distance

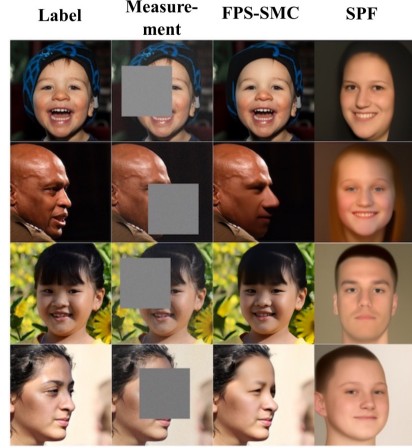

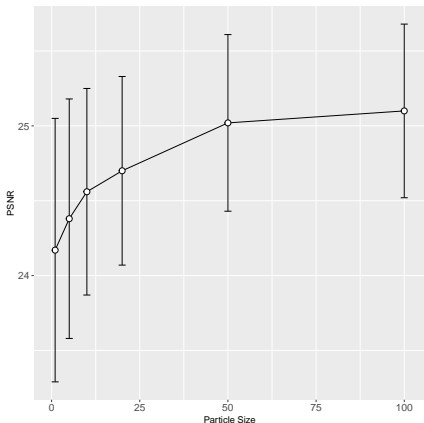

Figure 3: Comparison between FPS-SMC and Standard Particle Filtering (SPF) when particle size $M = 20$ on Inpainting (box) task over FFHQ dataset.

Figure 4: The relation between the particle size $M$ and the PSNR value, with $68\%$ ($\times 1$ std) confidence interval added

(FID), structural similarity index (SSIM), and Learned Perceptual Patch Similarity (LPIPS) (Zhang et al., 2018). These metrics enable a comprehensive assessment of image quality. All our experiments are carried out on a single A100 GPU.

## 5.2 EXPERIMENTAL RESULTS

In Tables 1 and 2, we compare FPS and FPS-SMC against various baselines in terms of FID and LPIPS. For additional quantitative results and comparisons in PSNR and SSIM, we relegate them to Tables 3 and 4 in Appendix E. The results demonstrate that FPS outperforms baselines significantly in tasks such as super resolution, inpainting (box), Gaussian deblurring, and motion deblurring. It is remarkable that in the challenging inpainting (random) task, where a substantial 92% of all pixels are masked out, FPS achieves the second-best performance. When assessing performance across the four metrics, FPS emerges as the front-runner in three of them, with superior results compared to the other baseline models.

## 5.3 ABLATION STUDY: INFLUENCE OF THE PARTICLE SIZE

In theory, we can solve the filtering problem with FPS-SMC more accurately when the particle size $M$ increases. By default, we apply $M = 20$ to all our experiments to make it computationally efficient. Comparing FPS (where $M = 1$) and FPS-SMC (where $M = 20$), we observe that FPS-SMC has an edge only with respect to the PSNR metric, but the difference in other three metrics SSIM, FID and LPIPS is small. To rigorously study the variation of image quality as a function of particle size $M$, we focus on inpainting (box) over FFHQ $256 \times 256$-1k validation set. As we can see in Fig. 4, PSNR metric increases as the particle size $M$ grows, but has diminishing returns once $M$ reaches 100. For the increasing running time of FPS-SMC for larger particle size $M$, we demonstrate in Appendix F.2 that it approximately holds that $t(M) \propto \sqrt{M}$. We also compare it with other popular methods, which shows that the running time of FPS-SMC is at the average level.

## 5.4 ABLATION STUDY: FPS-SMC VERSUS STANDARD PARTICLE FILTERING

In Eq. (12), we observe that

$$p_{\boldsymbol{\theta}}(\mathbf{x}_{k:N} \mid \mathbf{y}_{k:N}) \propto p_{\boldsymbol{\theta}}(\mathbf{y}_k \mid \mathbf{x}_k) \cdot p_{\boldsymbol{\theta}}(\mathbf{x}_k \mid \mathbf{x}_{k+1}) p_{\boldsymbol{\theta}}(\mathbf{x}_{k+1:N} \mid \mathbf{y}_{k+1:N}),$$

which leads to the standard way for implementing particle filtering. After obtaining $\{\mathbf{x}_{k+1}^{(j)}\}_{j \in [M]}$ from previous steps, we generate $M$ particles independently by simulating the unconditional backward process: $\overline{\mathbf{x}}_k^{(j)} \sim p_{\boldsymbol{\theta}}(\mathbf{x}_k \mid \mathbf{x}_{k+1}^{(j)})$ for $j \in [M]$. Afterwards, we randomly sample $M$ particles with replacement according to the distribution $\mathbb{P}\left[\mathbf{x}_k = \overline{\mathbf{x}}_k^{(j)}\right] \propto p_{\boldsymbol{\theta}}(\mathbf{y}_k \mid \overline{\mathbf{x}}_k^{(j)})$. We name it as Standard Particle Filtering (SPF). In theory, both FPS-SMC and SPF are consistent solvers for the Bayesian

filtering problem. However, in SPF, the $\{\mathbf{x}_k\}_{k=0}^N$ sequence can only get the measurement information from the re-sampling step, which provides weak signals for $\mathbf{x}_k$ to follow the measurement $\mathbf{y}$. As a result, we need a much larger particle size $M$ to obtain reasonable performance for SPF, especially in high dimensional spaces. In Fig. 3, we compare the performance between FPS-SMC and SPF when $M = 20$. As we can see, the particles generated by SPF are almost completely independent of the measurement. In contrast, we tailor the proposal distribution in FPS-SMC to significantly reduce the variance, resulting in an effective algorithm with a smaller particle size.

Table 1: Quantitative results (FID, LPIPS) of our model and existing models on a various of linear inverse problems on FFHQ $256 \times 256$-1k validation dataset.

| FFHQ | Super Resolution | | Inpainting (box) | | Gaussian Deblur | | Inpainting (random) | | Motion Deblur | |
|---|---|---|---|---|---|---|---|---|---|---|
| Methods | FID | LPIPS | FID | LPIPS | FID | LPIPS | FID | LPIPS | FID | LPIPS |
| **FPS** | 26.66 | 0.212 | **26.13** | **0.141** | 30.03 | **0.248** | 35.21 | 0.265 | 26.18 | **0.221** |
| **FPS-SMC** | **26.62** | **0.210** | 26.51 | 0.150 | **29.97** | 0.253 | 33.10 | 0.275 | **26.12** | 0.227 |
| DPS | 39.35 | 0.214 | 33.12 | 0.168 | 44.05 | 0.257 | **21.19** | **0.212** | 39.92 | 0.242 |
| DDRM | 62.15 | 0.294 | 42.93 | 0.204 | 74.92 | 0.332 | 69.71 | 0.587 | - | - |
| MCG | 87.64 | 0.520 | 40.11 | 0.309 | 101.2 | 0.340 | 29.26 | 0.286 | - | - |
| PnP-ADMM | 66.52 | 0.353 | 151.9 | 0.406 | 90.42 | 0.441 | 123.6 | 0.692 | - | - |
| Score-SDE | 96.72 | 0.563 | 60.06 | 0.331 | 109.0 | 0.403 | 76.54 | 0.612 | - | - |
| ADMM-TV | 110.6 | 0.428 | 68.94 | 0.322 | 186.7 | 0.507 | 181.5 | 0.463 | - | - |

Table 2: Quantitative results (FID, LPIPS) of our model and existing models on a various of linear inverse problems on ImageNet $256 \times 256$-1k validation dataset.

| ImageNet | Super Resolution | | Inpainting (box) | | Gaussian Deblur | | Inpainting (random) | | Motion Deblur | |
|---|---|---|---|---|---|---|---|---|---|---|
| Methods | FID | LPIPS | FID | LPIPS | FID | LPIPS | FID | LPIPS | FID | LPIPS |
| **FPS** | 47.32 | 0.329 | **33.19** | **0.204** | 54.41 | **0.396** | 42.68 | 0.325 | 52.22 | 0.370 |
| **FPS-SMC** | **47.30** | **0.316** | 33.24 | 0.212 | **54.21** | 0.403 | 42.77 | 0.328 | **52.16** | **0.365** |
| DPS | 50.66 | 0.337 | 38.82 | 0.262 | 62.72 | 0.444 | **35.87** | **0.303** | 56.08 | 0.389 |
| DDRM | 59.57 | 0.339 | 45.95 | 0.245 | 63.02 | 0.427 | 114.9 | 0.665 | - | - |
| MCG | 144.5 | 0.637 | 39.74 | 0.330 | 95.04 | 0.550 | 39.19 | 0.414 | - | - |
| PnP-ADMM | 97.27 | 0.433 | 78.24 | 0.367 | 100.6 | 0.519 | 114.7 | 0.677 | - | - |
| Score-SDE | 170.7 | 0.701 | 54.07 | 0.354 | 120.3 | 0.667 | 127.1 | 0.659 | - | - |
| ADMM-TV | 130.9 | 0.523 | 87.69 | 0.319 | 155.7 | 0.588 | 189.3 | 0.510 | - | - |

## 6 CONCLUSION

We introduce Filtering Posterior Sampling (FPS), a novel approach addressing the challenges inherent in diffusion models when solving linear inverse problems. FPS leverages a unique perspective on Bayesian posterior sampling, establishing an equivalence with Bayesian filtering and thereby enabling a transition from a sampling problem to a filtering one. Notably, our FPS algorithm employs particle filtering and sequential Monte Carlo techniques without necessitating resource-intensive optimizations or the training of supplementary neural networks. It offers compatibility with any Markovian sampler and showcases versatility across diverse linear inverse problems. The flexibility in compute-resource allocation in our method ensures an optimal balance between computational effort and sampling quality. On a theoretical front, we demonstrated the global convergence guarantee of our FPS method, marking a pioneering achievement in diffusion posterior sampling. Empirical evaluations across multiple computer vision tasks, including inpainting, super-resolution, and motion deblur further demonstrate the superiority of FPS over prevailing methods on benchmarks like FFHQ and ImageNet. This research paves the way for enhanced efficiency and precision in leveraging diffusion models for intricate linear inverse problems.

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

## A  MORE BACKGROUND ON CONTINUOUS DIFFUSION MODELS

**Continuous Diffusion Models**  In the continuous-time diffusion models (Song et al., 2020b), Gaussian noise is sequentially injected to data in the forward process and then we create samples by progressive denoising from noise in the backward process. Generally, the forward process follows a stochastic differential equation (SDE) with the following form:

$$d\mathbf{x}_t = \boldsymbol{\mu}(\mathbf{x}_t, t)dt + \sigma(t)d\mathbf{W}_t \quad \text{for } t \in [0, T]. \tag{14}$$

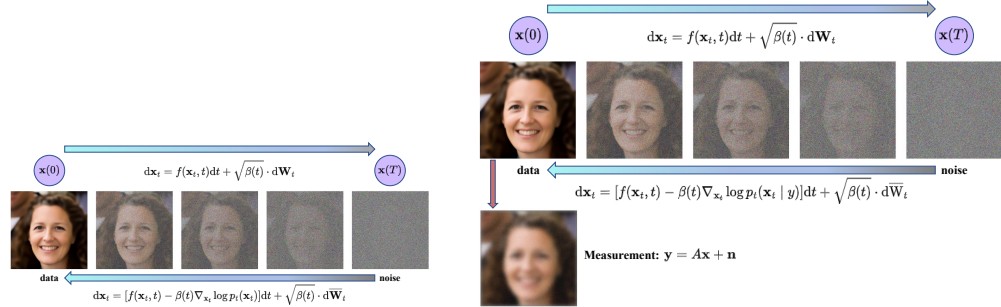

(a) Unconditional Generative Model       (b) Conditional Generative Model

Figure 5: Compared with the unconditional case, we replace the score function with the posterior score function in the backward process.

Here, $\mathbf{W}_t(\cdot)$ is the standard Brownian motion. $\boldsymbol{\mu}(\mathbf{x}_t, t)$ and $\sigma(t)$ are the drift term and the diffusion term respectively. The diffusion process starts from $\mathbf{x}(0) \sim p_{\text{data}}(\mathbf{x})$, the latent distribution of data, and $\mathbf{x}(T) \sim p_T(\mathbf{x})$, which is almost a pure Gaussian noise. As is stated in Anderson (1982), the reverse of the diffusion process (Eq. (14)) is also a diffusion process, with the formulation:

$$d\mathbf{x}_t = \left[ \boldsymbol{\mu}(\mathbf{x}_t, t) - \sigma(t)^2 \nabla_{\mathbf{x}_t} \log p_t(\mathbf{x}_t) \right] dt + \sigma(t)d\overline{\mathbf{W}}_t. \tag{15}$$

Here, $\overline{W}_t$ is a standard Brownian motion with time flows backwards from $T$ to 0. $\nabla_{\mathbf{x}_t} \log p_t(\mathbf{x}_t)$ is the score function of the noisy data distribution at time $t$, which is learned via denoising score matching (Vincent, 2011). It is worth mentioned that Eq. (15) is not the only backward diffusion process whose solution trajectories sampled at $t$ follow the distribution $p_t(\mathbf{x}_t)$. Practically, a widely used framework is the variance preserving SDE (VP-SDE) in Dhariwal & Nichol (2021):

$$d\mathbf{x}_t = -\frac{\beta(t)}{2}\mathbf{x}_t dt + \sqrt{\beta(t)}d\mathbf{W}_t. \tag{16}$$

Here $\beta(t) > 0$ is the noise schedule, which is usually chosen as a linear function over $t$. A standard Gaussian distribution can be achieved for $t = T$, i.e. $\mathbf{x}_T \sim \mathcal{N}(0, \boldsymbol{I})$. Its corresponding backward diffusion is:

$$d\mathbf{x}_t = \left[ -\frac{\beta(t)}{2}\mathbf{x}_t - \beta(t)\nabla_{\mathbf{x}_t} \log p_t(\mathbf{x}_t) \right] dt + \sqrt{\beta(t)}d\overline{\mathbf{W}}_t \tag{17}$$

**Conditional Diffusion Models**  For most existing conditional diffusion models, they analyze the following SDE

$$d\mathbf{x}_t = \left[ -\frac{\beta(t)}{2}\mathbf{x}_t - \beta(t)\nabla_{\mathbf{x}_t} \log p_t(\mathbf{x}_t \mid \mathbf{y}) \right] dt + \sqrt{\beta(t)}d\overline{\mathbf{W}}_t \tag{18}$$

where the score function $\nabla_{\mathbf{x}_t} \log p_t(\mathbf{x}_t)$ is replaced by the conditional score function $\nabla_{\mathbf{x}_t} \log p_t(\mathbf{x}_t \mid \mathbf{y})$. The key problem falls onto the estimation of conditional score function without re-training. According to the Bayes rule, it holds that:

$$\nabla_{\mathbf{x}_t} \log p_t(\mathbf{x}_t|\mathbf{y}) = \nabla_{\mathbf{x}_t} \log p_t(\mathbf{x}_t) + \nabla_{\mathbf{x}_t} \log p(\mathbf{y}|\mathbf{x}_t).$$

The first term is the unconditional score function, which can be estimated by our pretrained score estimator. The second term does not have an explicit form even in the linear inverse problems. In the

projection-type approaches such as Song et al. (2020b); Chung et al. (2022c); Chung & Ye (2022); **?**, the authors consider the noiseless case where $\mathbf{n} \approx 0$. After an unconditional update, they ignore the posterior score function $\nabla_{\mathbf{x}_t} \log p(\mathbf{y}|\mathbf{x}_t)$. Instead, they conduct a projection step to keep the model consistency $\mathbf{y} = A\mathbf{x}$ still holds. SVD-type approaches Kawar et al. (2022) use SVD to simplify the linear inverse problem to a component-wise unconditional diffusion model. Besides, another line of works such as Chung et al. (2022a); Kawar et al. (2021); Bardsley et al. (2014); Song et al. (2023a); Jalal et al. (2021) try different ways to provide a close approximation of $\nabla_{\mathbf{x}_t} \log p(\mathbf{y}|\mathbf{x}_t)$. Their main idea is to use $\nabla_{\mathbf{x}_t} \log p_t(\mathbf{y} \mid \mathbf{x}_t) \approx \nabla_{\mathbf{x}_t} \log p_t(\mathbf{y} \mid \hat{\mathbf{x}}_0(\mathbf{x}_t))$ to approximate the conditional score with another explicit score. Jalal et al. (2021) utilize the approximation $\nabla_{\mathbf{x}_t} \log p_t(\mathbf{y} \mid \mathbf{x}_t) \approx \frac{A^\top(\mathbf{y}-A\mathbf{x}_t)}{\sigma^2+\gamma_t^2}$ where $\{\gamma_t\}$ are hyper-parameters. Chung et al. (2022a) explicitly compute $\nabla_{\mathbf{x}_t} \log p_t(\mathbf{y} \mid \hat{\mathbf{x}}_0(\mathbf{x}_t))$, based on which Song et al. (2023a) propose $p_t(\mathbf{y} \mid \mathbf{x}_t) \approx \mathcal{N}(A\hat{\mathbf{x}}_0(\mathbf{x}_t), r_t^2 AA^\top + \sigma^2 I)$. Here,

$$\hat{\mathbf{x}}_0(\mathbf{x}_t) = \mathbb{E}[\mathbf{x}_0 \mid \mathbf{x}_t] = \mathbf{x}_t + \sigma_t^2 \nabla_{\mathbf{x}_t} \log p_t(\mathbf{x}_t)$$

is the score-based estimated initial data point $\mathbf{x}_0$ given its noisy version $\mathbf{x}_t$. Besides the mean $\mathbb{E}[\mathbf{x}_0 \mid \mathbf{x}_t]$, its variance is additionally considered in Boys et al. (2023). Another way to approximate the conditional score is directly tuning the score network (i.e. U-Net) over $y$, which is recently proposed and implemented by Barbano et al. (2023); Chung et al. (2023).

Instead of approximating the conditional score function at each time step, we use a completely different framework and apply Bayesian filtering model to approximate the posterior sampling directly. This method is computationally efficient since we do not need to take derivative on estimated score function.

**Monte Carlo based Diffusion Models and Asymptotic Consistency** A number of results are proposed recently on Monte Carlo based conditional diffusion models, including several concurrent works. Trippe et al. (2022) developed an efficient method SMCDiff to sample scaffolds from a distribution conditioned on a given motif, which is a popular and pivotal task in protein design. This prior work only considered the inpainting-type task, which is just a special type of linear inverse problem. Besides, the proposed asymptotic consistency requires the perfect match between forward and backward unconditional diffusion process, just like our Proposition 4.1. Based on this work, Wu et al. (2023) introduced the Twisted Diffusion Sampler (TDS), a sequential Monte Carlo (SMC) algorithm on conditional diffusion models as well. By using a constant $\{\mathbf{y}_k\}_{k=0}^N$ sequence, there is no explicit formulation on $p(\mathbf{x}_k \mid \mathbf{x}_{k+1}, \mathbf{y}_k)$ for $\mathbf{y}_k = \mathbf{y}_0$. A tractable twisting function and Monte Carlo based technique are needed to approximate the probability. For the asymptotic consistency, several regularity conditions are required on the twisting function sequence. As a simultaneous work, we use different methods to solve the same problem. By introducing an appropriately designed $\{\mathbf{y}_k\}$ sequence, we transform the posterior sampling problem into a Bayesian filtering problem where all transition probabilities are explicit, which can be elegantly solved by variance reduced Sequential Monte Carlo with asymptotic consistency holds. Another concurrent work proposed by Cardoso et al. (2023) introduces Monte Carlo guided diffusion (MCGdiff) to solve Bayesian linear inverse problems. Compared to Wu et al. (2023), a different proposal kernel is used on SMC.

**Formulations of DDPM and DDIM** In this part, we still use the notation $\{\beta_k\}_{k=1}^N$, $\{\alpha_k\}_{k=1}^N$ and $\{\overline{\alpha}_k\}_{k=1}^N$ introduced in Section 2. For the Denoising Diffusion Probabilistic Model (DDPM), the forward process is Markovian with

$$q(\mathbf{x}_{1:N} \mid \mathbf{x}_0) = \prod_{k=1}^N q(\mathbf{x}_k \mid \mathbf{x}_{k-1}), \text{ where } q(\mathbf{x}_k \mid \mathbf{x}_{k-1}) = \mathcal{N}\left(\sqrt{\alpha_k}\mathbf{x}_{k-1}, \beta_k^2 \boldsymbol{I}\right).$$

A direct corollary for the forward process is that $q(\mathbf{x}_k \mid \mathbf{x}_0) = \mathcal{N}\left(\sqrt{\overline{\alpha}_k}\mathbf{x}_0, (1-\overline{\alpha}_k)\boldsymbol{I}\right)$, which leads to the following expression for $\mathbf{x}_k$:

$$\mathbf{x}_k = \sqrt{\overline{\alpha}_k}\mathbf{x}_0 + \sqrt{1-\overline{\alpha}_k} \cdot \boldsymbol{\varepsilon} \text{ where } \boldsymbol{\varepsilon} \sim \mathcal{N}(0, \boldsymbol{I}).$$

With $\mathbf{x}_0$ added to the transition $q(\mathbf{x}_k \mid \mathbf{x}_{k-1}, \mathbf{x}_0)$, we can keep the marginals unchanged but the forward process not Markovian any more. For the backward process proposed in Denoising Diffusion Implicit Model (DDIM) (Song et al., 2020a), we fix the prior $p_{\boldsymbol{\theta}}(\mathbf{x}_T) = \mathcal{N}(0, I)$ and we can generate $\mathbf{x}_{k-1}$ from $\mathbf{x}_k$ via:

$$\mathbf{x}_{k-1} = \sqrt{\overline{\alpha}_{k-1}} \cdot \left(\frac{\mathbf{x}_k - \sqrt{1-\overline{\alpha}_k}\boldsymbol{\varepsilon}_{\boldsymbol{\theta}}^k(\mathbf{x}_k)}{\sqrt{\overline{\alpha}_k}}\right) + \sqrt{1-\overline{\alpha}_{k-1}-\sigma_k^2} \cdot \boldsymbol{\varepsilon}_{\boldsymbol{\theta}}^k(\mathbf{x}_k) + \sigma_k \boldsymbol{\varepsilon}_k.$$

Here, $\varepsilon_k \sim \mathcal{N}(0, \boldsymbol{I})$ is a standard Gaussian noise independent of $\mathbf{x}_k$ and

$$\varepsilon_{\boldsymbol{\theta}}^k(\mathbf{x}_k) \approx -\nabla_{\mathbf{x}_k} \log p_k(\mathbf{x}_k)/\sqrt{1 - \overline{\alpha}_k}$$

is an approximated sample-dependent noise learned by score matching. Noise that different choices of variance $\sigma_k$ leads to different backward processes but all of them keep the trajectory with the same marginal distributions $p_k(\mathbf{x}_k)$. When we choose $\sigma_k^2 = \beta_k \cdot \frac{1 - \overline{\alpha}_{k-1}}{1 - \overline{\alpha}_k}$ for all $k$, the process degrades to the case of DDPM.

## B  EXPLICIT FORMULATION OF FPS WITH DDIM PLUGGED IN

As a compliment of our framework described in the main text, we make explicit computation in this section for FPS with DDIM plugged in. First, we introduce the duplex forward diffusion and quantitatively show the connection between posterior sampling and the Bayesian filtering problem.

### B.1  DUPLEX FORWARD DIFFUSION PROCESS

For the discrete case of diffusion models such as DDPM and DDIM, their forward diffusion can be expressed as the following recursive linear Gaussian process:

$$\mathbf{x}_k = \sqrt{1 - \beta_k} \cdot \mathbf{x}_{k-1} + \sqrt{\beta_k}\mathbf{z}_k \tag{19}$$

where $\mathbf{z}_k \sim \mathcal{N}(\mathbf{0}, \boldsymbol{I})$ is a standard Gaussian noise independent with $\mathbf{x}_{k-1}$. Meanwhile, we make:

$$\mathbf{y}_k = \sqrt{1 - \beta_k} \cdot \mathbf{y}_{k-1} + \sqrt{\beta_k}A\mathbf{z}_k. \tag{20}$$

as the forward process for measurements where $\mathbf{y}_0 = \mathbf{y}$ is the initial measurement. Notice that these two $\{\mathbf{z}_k\}$ represent the exactly same noise vector. So, if we combine data and measurement as a pair, the forward SDE can be defined in the following form:

$$\mathrm{d}\begin{pmatrix}\mathbf{x}_t \\ \mathbf{y}_t\end{pmatrix} = -\frac{\beta(t)}{2}\begin{pmatrix}\mathbf{x}_t \\ \mathbf{y}_t\end{pmatrix}\mathrm{d}t + \sqrt{\beta(t)}\begin{pmatrix}\boldsymbol{I}_D & 0 \\ 0 & \boldsymbol{A}\end{pmatrix}\mathrm{d}\mathbf{W}_t.$$

Here, $\mathbf{W}_t$ is the standard $2D$-dimensional Wiener process. After simple calculations over Equations (19), (20), we can directly build connection between $\mathbf{x}_0, \mathbf{y}_0$ and $\mathbf{x}_t, \mathbf{y}_t$.

**Proposition B.1.** *For the case of duplex forward diffusion process, we can simply represent it by:*

$$\mathbf{x}_k = \sqrt{\overline{\alpha}_k} \cdot \mathbf{x}_0 + \sqrt{1 - \overline{\alpha}_k} \cdot \mathbf{z}, \quad \text{and } \mathbf{y}_k = \sqrt{\overline{\alpha}_k} \cdot \mathbf{y}_0 + \sqrt{1 - \overline{\alpha}_k} \cdot \boldsymbol{A}\mathbf{z}, \quad \mathbf{z} \sim \mathcal{N}(\mathbf{0}, \boldsymbol{I}_D).$$

*We can also conclude that:*

$$q(\mathbf{y}_k \mid \mathbf{x}_k) = \mathcal{N}(\boldsymbol{A}\mathbf{x}_k, \ \sigma^2\overline{\alpha}_k \cdot \boldsymbol{I}_d),$$

*which means $\mathbf{y}_k$ also follows a normal distribution with mean $\boldsymbol{A}\mathbf{x}_k$ and a decreasing variance $\sigma^2\overline{\alpha}_k$. For the initial step $k = 0$, we have $\overline{\alpha}_0 = 1$ and it degrades to the model consistency function $\mathbf{y}_0 = \mathbf{y} = \boldsymbol{A}\mathbf{x}_0 + \mathbf{n}$ where $\mathbf{n} \sim \mathcal{N}(\mathbf{0}, \sigma^2\boldsymbol{I})$.*

Unlike $\mathbf{x}_0$ which we can only estimate, we do know $\mathbf{y}_0$, the initial measurement without any extra noise, which makes the score function for $\mathbf{y}_t$ tractable.

**Proposition B.2.** *For the case of duplex forward diffusion process, the score function of $\mathbf{y}_k$ is:*

$$\nabla_{\mathbf{y}_k} \log q_k(\mathbf{y}_k) = -\frac{1}{1 - \overline{\alpha}_k}(\boldsymbol{A}\boldsymbol{A}^\top)^{-1} \cdot \left(\mathbf{y}_k - \sqrt{\overline{\alpha}_k}\mathbf{y}_0\right).$$

According to Equation (20), the distribution of sequence $\{\mathbf{y}_k\}$ given the measurement $\mathbf{y} = \mathbf{y}_0$ is tractable since the diffusion sampling is a Markov chain:

$$q(\mathbf{y}_{1:N} \mid \mathbf{y}_0) = \prod_{k=1}^{N} q(\mathbf{y}_k \mid \mathbf{y}_{k-1}).$$

For conditional sampling, we do not know the distribution of $\{\mathbf{x}_k\}_{k=0}^N$ but know that of $\{\mathbf{y}_k\}_{k=0}^N$. Therefore, calculating the hidden Markov chain from the $\{\mathbf{y}_k\}_{k=0}^N$ sequence is equivalent to solving a reverse-time Bayesian filtering problem since we have access to $p_{\boldsymbol{\theta}}(\mathbf{y}_k \mid \mathbf{x}_k) = q(\mathbf{y}_k \mid \mathbf{x}_k)$ from Proposition B.1 and $p_{\boldsymbol{\theta}}(\mathbf{x}_{k-1} \mid \mathbf{x}_k)$ from score-based backward diffusion process.

## B.2 BACKWARD PROCESS OF FILTERING POSTERIOR SAMPLING

**Step 1: Generating Sequence** $\{\mathbf{y}_k\}_{k=0}^N$  For the DDIM backward process with regard to $\{\mathbf{y}_k\}_{k=0}^N$, we have:

$$\mathbf{y}_{k-1} = \sqrt{\overline{\alpha}_{k-1}}\hat{\mathbf{y}}_0 + \sqrt{\frac{(1-c)(1-\overline{\alpha}_{k-1})}{1-\overline{\alpha}_k}}(\mathbf{y}_k - \overline{\alpha}_k\hat{\mathbf{y}}_0) + \sqrt{c(1-\overline{\alpha}_{k-1})} \cdot \mathbf{A}\varepsilon.$$

where $\varepsilon \sim \mathcal{N}(0, I)$ and $\hat{\mathbf{y}}_0 = \mathbb{E}[\mathbf{y}_0 \mid \mathbf{y}_k]$ is a posterior mean calculated by Tweedie's formula (Efron, 2011; Kim & Ye, 2021). Here, $\hat{\mathbf{y}}_0 = \mathbf{y}_0$ since the ground truth measurement $\mathbf{y}_0$ is given and no estimation or approximation is needed.

$$\mathbf{y}_{k-1} = \sqrt{\overline{\alpha}_{k-1}}\mathbf{y}_0 + \sqrt{\frac{(1-c)(1-\overline{\alpha}_{k-1})}{1-\overline{\alpha}_k}}(\mathbf{y}_k - \overline{\alpha}_k\mathbf{y}_0) + \sqrt{c(1-\overline{\alpha}_{k-1})} \cdot \mathbf{A}\varepsilon. \tag{21}$$

For $\mathbf{y}_N$, we have already proved that $q(\mathbf{y}_N \mid \mathbf{y}_0) = \mathcal{N}(\sqrt{\overline{\alpha}_N}\mathbf{y}_0, (1-\overline{\alpha}_N)\mathbf{A}^\top \mathbf{A})$ in Proposition B.1. Since $\overline{\alpha}_N$ converges to 0 for large enough $N$, therefore we can directly make the following sampling:

$$\mathbf{y}_N = \mathbf{A}\varepsilon_N \quad \text{for } \varepsilon_N \sim \mathcal{N}(0, \mathbf{I}). \tag{22}$$

**Remark B.1.** *Since $\mathbf{y}_0$ is given in the case of linear inverse problems, it is also reasonable to directly use the forward process of sequence $\{\mathbf{y}_k\}_{k=0}^N$ where $\mathbf{y}_k = \sqrt{1-\beta_k} \cdot \mathbf{y}_{k-1} + \sqrt{\beta_k}\mathbf{A}\mathbf{z}_k$ for $\mathbf{z}_k \sim \mathcal{N}(\mathbf{0}, \mathbf{I})$ since both the forward and backward sequences have exactly the same marginal distributions $q_k(\mathbf{y}_k)$. In this paper, we use the backward process to make our model structurally aligned.*

**Step 2: Generating Backward Sequence of** $\{\mathbf{x}_k\}_{k=0}^N$  Given the $\{\mathbf{y}_k\}_{k=0}^N$ sequence given, we continue generate the $\{\mathbf{x}_k\}_{k=0}^N$ backward sequence by using the filtering method. In this section, we provide the explicit Gaussian formulation of each $p_{\boldsymbol{\theta}}(\mathbf{x}_{k-1} \mid \mathbf{x}_k, \mathbf{y}_{k-1})$ by using Eq. (9), which depends on the unconditional update model

$$p_{\boldsymbol{\theta}}(\mathbf{x}_{k-1} \mid \mathbf{x}_k) := \mathcal{N}\left(\boldsymbol{\mu}_k(\mathbf{x}_k, \boldsymbol{\theta}), \Sigma_k\right)$$

we choose to plug in.

For example, for DDPM framework, we have:

$$\boldsymbol{\mu}_k(\mathbf{x}_k, \boldsymbol{\theta}) = \frac{1}{\sqrt{\alpha_k}}\mathbf{x}_k + \frac{\beta_k}{\sqrt{\alpha_k}} \cdot \boldsymbol{s}_{\boldsymbol{\theta}}(\mathbf{x}_k, t_k) \quad \text{and} \quad \Sigma_k = \frac{\beta_k(1-\overline{\alpha}_{k-1})}{1-\overline{\alpha}_k} \cdot \boldsymbol{I},$$

where $\boldsymbol{s}_{\boldsymbol{\theta}}(\mathbf{x}_k, t_k)$ is the pretrained approximation for the score function $\nabla_{\mathbf{x}_k} \log p_k(\mathbf{x}_k)$.

For the DDIM framework, we have a more general formulation with:

$$\boldsymbol{\mu}_k(\mathbf{x}_k, \boldsymbol{\theta}) = \frac{1}{\alpha_k} \cdot (\mathbf{x}_k + (1-\overline{\alpha}_k)\boldsymbol{s}_{\boldsymbol{\theta}}(\mathbf{x}_k, t_k)) - \sqrt{(1-\overline{\alpha}_{k-1} - \sigma_k^2) \cdot (1-\overline{\alpha}_k)} \cdot \boldsymbol{s}_{\boldsymbol{\theta}}(\mathbf{x}_k, t_k).$$

and

$$\Sigma_k = \sigma_k^2 \cdot \boldsymbol{I}.$$

We have claimed that when $\sigma_k^2 = \frac{\beta_k(1-\overline{\alpha}_{k-1})}{1-\overline{\alpha}_k}$, the DDIM framework becomes a DDPM. Therefore, we apply the more general DDIM framework and make $\sigma_k^2 = c(1-\overline{\alpha}_{k-1})$ with $c \in [0, 1]$ left as a tunable hyper-parameter, which makes:

$$\boldsymbol{\mu}_k^{\text{DDIM}}(\mathbf{x}_k, \boldsymbol{\theta}) = \frac{1}{\alpha_k} \cdot (\mathbf{x}_k + (1-\overline{\alpha}_k)\boldsymbol{s}_{\boldsymbol{\theta}}(\mathbf{x}_k, t_k)) - \sqrt{(1-c)(1-\overline{\alpha}_{k-1}) \cdot (1-\overline{\alpha}_k)} \cdot \boldsymbol{s}_{\boldsymbol{\theta}}(\mathbf{x}_k, t_k) \tag{23}$$

and the covariance matrix

$$\Sigma_k^{\text{DDIM}} = c(1-\overline{\alpha}_{k-1}) \cdot \boldsymbol{I}. \tag{24}$$

After combining $p_{\boldsymbol{\theta}}(\mathbf{y}_{k-1} \mid \mathbf{x}_{k-1})$ and $p_{\boldsymbol{\theta}}(\mathbf{x}_{k-1} \mid \mathbf{x}_k)$, the following conclusion can be obtained by linear filtering.

**Proposition B.3.** *Under the case that*

$$p_{\boldsymbol{\theta}}(\mathbf{y}_{k-1} \mid \mathbf{x}_{k-1}) = \mathcal{N}(\boldsymbol{A}\mathbf{x}_{k-1}, \, \sigma^2 \overline{\alpha}_{k-1} \cdot \boldsymbol{I}) \quad and \quad p_{\boldsymbol{\theta}}(\mathbf{x}_{k-1} \mid \mathbf{x}_k) = \mathcal{N}\left(\boldsymbol{\mu}_k^{\mathrm{DDIM}}(\mathbf{x}_k, \boldsymbol{\theta}), \, \Sigma_k^{\mathrm{DDIM}}\right)$$

*with* $\boldsymbol{\mu}_k^{\mathrm{DDIM}}(\mathbf{x}_k, \boldsymbol{\theta})$ *and* $\Sigma_k^{\mathrm{DDIM}}$ *defined in Equations (23), (24). We can conclude that:*

$$p_{\boldsymbol{\theta}}(\mathbf{x}_{k-1} \mid \mathbf{x}_k, \mathbf{y}_{k-1}) = \mathcal{N}(\boldsymbol{\mu}_k^{\mathrm{FPS}}(\mathbf{x}_k, \mathbf{y}_{k-1}, \boldsymbol{\theta}), \, \Sigma_k^{\mathrm{FPS}})$$

*where*

$$\Sigma_k^{\mathrm{FPS}} = \left( \left(\Sigma_k^{\mathrm{DDIM}}\right)^{-1} + \frac{1}{\sigma^2 \cdot \overline{\alpha}_{k-1}} \cdot \boldsymbol{A}^\top \boldsymbol{A} \right)^{-1}$$

*and*

$$\boldsymbol{\mu}_k^{\mathrm{FPS}}(\mathbf{x}_k, \mathbf{y}_{k-1}, \boldsymbol{\theta}) = \Sigma_k^{\mathrm{FPS}} \cdot \left( \left(\Sigma_k^{\mathrm{DDIM}}\right)^{-1} \boldsymbol{\mu}_k^{\mathrm{DDIM}}(\mathbf{x}_k, \boldsymbol{\theta}) + \frac{1}{\sigma^2 \cdot \overline{\alpha}_{k-1}} \cdot \boldsymbol{A}^\top \mathbf{y}_{k-1} \right).$$

Till now, we have figured out all the details of FPS as well as its variant FPS-SMC. We rewrite the pseudo-algorithms of FPS-SMC with DDIM plugged in as the following Algorithm 2. When the particle size $M = 1$, it degrades to FPS.

---

**Algorithm 2** The Framework of FPS-SMC with DDIM Plugged-in

---

1: Given: $N, \mathbf{y}, \{\overline{\alpha}_k\}_{k \in [N]}, c$, particle size $M$ and the score estimator $\boldsymbol{s}_{\boldsymbol{\theta}}(\cdot, \cdot)$.
2: $\mathbf{y}_N = \boldsymbol{A}\varepsilon_N, \mathbf{x}_N^{(j)} = \varepsilon_N \quad \varepsilon_N \sim \mathcal{N}(0, I)$ for $j \in [M]$.
3: **for** $k = N, N-1, \ldots, 1$ **do**
4:     $p_k \leftarrow \sqrt{\frac{(1-c)(1-\overline{\alpha}_{k-1})}{1-\overline{\alpha}_k}}, q_k \leftarrow \sqrt{c(1-\overline{\alpha}_{k-1})}$
5:     $\mathbf{y}_{k-1} \leftarrow \sqrt{\overline{\alpha}_{k-1}}\mathbf{y} + p_k(\mathbf{y}_k - \sqrt{\overline{\alpha}_k}\mathbf{y}) + q_k \boldsymbol{A}\varepsilon_i$
6: **end for**
7: **for** $k = N-1, \ldots, 0$ **do**
8:     Sample $\overline{\mathbf{x}}_k^{(j)} \sim \mathcal{N}(\boldsymbol{\mu}_{k+1}^{\mathrm{FPS}}(\mathbf{x}_{k+1}^{(j)}, \mathbf{y}_k, \hat{\boldsymbol{\theta}}), \Sigma_{k+1}^{\mathrm{FPS}})$ for $j = 1, 2, \ldots, M$
      where $\boldsymbol{\mu}_{k+1}^{\mathrm{FPS}}(\mathbf{x}_{k+1}, \mathbf{y}_k, \hat{\boldsymbol{\theta}}), \Sigma_{k+1}^{\mathrm{FPS}}$ are defined in Proposition B.3.
9:     Pick $M$ samples with replacement from $\left\{\overline{\mathbf{x}}_k^{(j)}\right\}_{j \in [M]}$ with probability distribution (13). We
      denote the sample set as $\left\{\mathbf{x}_k^{(j)}\right\}_{j \in [M]}$.
10: **end for**
11: The output $\mathbf{x}_0$ is uniformly sampled from $\{\mathbf{x}_0^{(j)}\}_{j \in [M]}$.

---

## C    PROOF OF PROPOSITIONS IN APPENDIX B

In this section, we provide detailed proofs for the three propositions stated in Section 3.

### C.1    PROOF OF PROPOSITION B.1

According to the duplex forward diffusion framework,

$$\mathbf{x}_k = \sqrt{\overline{\alpha}_k} \cdot \mathbf{x}_0 + \sqrt{1 - \overline{\alpha}_k}\mathbf{z}, \quad \mathbf{y}_k = \sqrt{\overline{\alpha}_k} \cdot \mathbf{y}_0 + \sqrt{1 - \overline{\alpha}_k}\boldsymbol{A}\mathbf{z}.$$

We have:

$$\mathbf{y}_k - \boldsymbol{A}\mathbf{x}_k = \sqrt{\overline{\alpha}_k} \cdot (\mathbf{y}_0 - \boldsymbol{A}\mathbf{x}_0).$$

Since $\mathbf{y}_0 - \boldsymbol{A}\mathbf{x}_0 \sim \mathcal{N}(0, \sigma^2 \cdot \boldsymbol{I}_d)$, we can easily conclude that:

$$\mathbf{y}_k - \boldsymbol{A}\mathbf{x}_k \sim \mathcal{N}(\mathbf{0}, \sigma^2 \overline{\alpha}_k \cdot \boldsymbol{I}_d),$$

which is equivalent to:

$$\mathbf{y}_k \sim \mathcal{N}(\boldsymbol{A}\mathbf{x}_k, \sigma^2 \overline{\alpha}_k \cdot \boldsymbol{I}_d).$$

## C.2 PROOF OF PROPOSITION B.2

Given the fact that
$$\mathbf{y}_k \sim \mathcal{N}(\sqrt{\overline{\alpha}_k}\mathbf{y}_0, (1-\overline{\alpha}_k) \cdot \boldsymbol{A}\boldsymbol{A}^\top),$$
the density function as well as the score function can be explicitly written as:
$$q_k(\mathbf{y}_k) \propto \exp\left(-\frac{1}{2}(\mathbf{y}_k - \overline{\alpha}_k\mathbf{y}_0)^\top \frac{(\boldsymbol{A}\boldsymbol{A}^\top)^{-1}}{1-\overline{\alpha}_k} \cdot (\mathbf{y}_k - \overline{\alpha}_k\mathbf{y}_0)\right),$$

and
$$\nabla_{\mathbf{y}_k} \log q_k(\mathbf{y}_k) = -\frac{1}{1-\overline{\alpha}_k}(\boldsymbol{A}\boldsymbol{A}^\top)^{-1} \cdot (\mathbf{y}_k - \overline{\alpha}_k\mathbf{y}_0),$$

which comes to our conclusion.

## C.3 PROOF OF PROPOSITION B.3

For simplicity, we denote
$$p_{\boldsymbol{\theta}}(\mathbf{y}_{k-1} \mid \mathbf{x}_{k-1}) = \mathcal{N}(\tau_1, \Sigma_1) \quad \text{and} \quad p_{\boldsymbol{\theta}}(\mathbf{x}_{k-1} \mid \mathbf{x}_k) = \mathcal{N}(\tau_2, \Sigma_2),$$
where $\tau_1 = \boldsymbol{A}\mathbf{x}_{k-1}, \tau_2 = \boldsymbol{\mu}_k^{\text{DDIM}}(\mathbf{x}_k, \boldsymbol{\theta})$ and $\Sigma_1 = \sigma^2\overline{\alpha}_{k-1} \cdot \boldsymbol{I}, \Sigma_2 = \Sigma_k^{\text{DDIM}}$. Then,
$$p_{\boldsymbol{\theta}}(\mathbf{x}_{k-1} \mid \mathbf{x}_k, \mathbf{y}_{k-1}) \propto p_{\boldsymbol{\theta}}(\mathbf{x}_{k-1}, \mathbf{x}_k, \mathbf{y}_{k-1}) = p_{\boldsymbol{\theta}}(\mathbf{x}_{k-1} \mid \mathbf{x}_k) \cdot p_{\boldsymbol{\theta}}(\mathbf{y}_{k-1} \mid \mathbf{x}_{k-1}) \cdot p_{\boldsymbol{\theta}}(\mathbf{x}_k).$$

Here, we use the fact that $\mathbf{y}_{k-1}$ and $\mathbf{x}_k$ are independent given $\mathbf{x}_{k-1}$. Therefore:
$$p_{\boldsymbol{\theta}}(\mathbf{x}_{k-1} \mid \mathbf{x}_k, \mathbf{y}_{k-1}) \propto p_{\boldsymbol{\theta}}(\mathbf{x}_{k-1} \mid \mathbf{x}_k) \cdot p_{\boldsymbol{\theta}}(\mathbf{y}_{k-1} \mid \mathbf{x}_{k-1}),$$

which leads to:
$$p_{\boldsymbol{\theta}}(\mathbf{x}_{k-1} \mid \mathbf{x}_k, \mathbf{y}_{k-1}) \propto \exp\left(-\frac{1}{2}(\mathbf{x}_{k-1} - \tau_2)^\top \Sigma_2^{-1} \cdot (\mathbf{x}_{k-1} - \tau_2) - \frac{1}{2}(\mathbf{y}_{k-1} - \tau_1)^\top \Sigma_1^{-1} \cdot (\mathbf{y}_{k-1} - \tau_1)\right)$$

$$\propto \exp\left(-\frac{1}{2}\mathbf{x}_{k-1}^\top \Sigma_2^{-1} \cdot \mathbf{x}_{k-1} + \tau_2^\top \Sigma_2^{-1} \cdot \mathbf{x}_{k-1} - \frac{1}{2}\mathbf{x}_{k-1}^\top \boldsymbol{A}^\top \Sigma_1^{-1} \boldsymbol{A}\mathbf{x}_{k-1} + \mathbf{y}_{k-1}^\top \Sigma_1^{-1} \cdot \boldsymbol{A}\mathbf{x}_{k-1}\right)$$

$$= \exp\left(-\frac{1}{2}\mathbf{x}_{k-1}^\top (\Sigma_2^{-1} + \boldsymbol{A}^\top \Sigma_1^{-1}\boldsymbol{A}) \cdot \mathbf{x}_{k-1} + (\Sigma_2^{-1}\tau_2 + \boldsymbol{A}^\top \Sigma_1^{-1}\mathbf{y}_{k-1})^\top \mathbf{x}_{k-1}\right)$$

Therefore, we have $p_{\boldsymbol{\theta}}(\mathbf{x}_{k-1} \mid \mathbf{x}_k, \mathbf{y}_{k-1}) = \mathcal{N}(\boldsymbol{\mu}_k^{\text{FPS}}(\mathbf{x}_k, \mathbf{y}_{k-1}, \boldsymbol{\theta}), \Sigma_k^{\text{FPS}})$ where

$$\Sigma_k^{\text{FPS}} = (\Sigma_2^{-1} + \boldsymbol{A}^\top \Sigma_1^{-1}\boldsymbol{A})^{-1} = \left((\Sigma_k^{\text{DDIM}})^{-1} + \frac{1}{\sigma^2\overline{\alpha}_{k-1}}\boldsymbol{A}^\top\boldsymbol{A}\right)^{-1},$$

$$\boldsymbol{\mu}_k^{\text{FPS}}(\mathbf{x}_k, \mathbf{y}_{k-1}, \boldsymbol{\theta}) = \Sigma_k^{\text{FPS}} \cdot (\Sigma_2^{-1}\tau_2 + \boldsymbol{A}^\top \Sigma_1^{-1}\mathbf{y}_{i-1})$$

$$= \Sigma_k^{\text{FPS}} \cdot \left((\Sigma_k^{\text{DDIM}})^{-1}\boldsymbol{\mu}_k^{\text{DDIM}}(\mathbf{x}_k, \boldsymbol{\theta}) + \frac{1}{\sigma^2 \cdot \overline{\alpha}_{k-1}} \cdot \boldsymbol{A}^\top\mathbf{y}_{k-1}\right),$$

which comes to our conclusion.

# D THEORETICAL UNDERSTANDING OF FPS AND FPS-SMC

## D.1 THE CONTINUOUS LIMIT OF FPS

In this section, we are going to reveal the theoretical properties of Filtering Posterior Sampling (FPS) in the continuous setting where the time step $\Delta t \to 0$. In the Diffusion Posterior Sampling (DPS), the backward diffusion process has the following formulation:

$$\mathrm{d}\mathbf{x}_t = \left[-\frac{\beta(t)}{2}\mathbf{x}_t - \beta(t)\nabla_{\mathbf{x}_t} \log p_t(\mathbf{x}_t \mid \mathbf{y})\right] \mathrm{d}t + \sqrt{\beta(t)}\mathrm{d}\overline{\mathbf{W}}_t.$$

Actually, when using diffusion models to solve inverse problem or posterior sampling, the SDE above is the only framework to follow. The only difference is how to approximate $\nabla_{\mathbf{x}_t} \log p_t(\mathbf{x}_t \mid \mathbf{y})$. Here, we are going to theoretically answer the following question:

For the continuous FPS, what is our actual approximation for $\nabla_{\mathbf{x}_t} \log p_t(\mathbf{x}_t \mid \mathbf{y})$?

Notice that when $\Delta t \to 0$, $\boldsymbol{\mu}_t^{\text{DDIM}}$ and $\Sigma_t^{\text{DDIM}}$ have the following form:

$$\Sigma_t^{\text{DDIM}} = \beta(t)\Delta t \cdot \boldsymbol{I}, \;\; \boldsymbol{\mu}_t^{\text{DDIM}}(\mathbf{x}_t) = \mathbf{x}_t + \left(\frac{\beta(t)}{2}\mathbf{x}_t + \beta(t)\nabla_{\mathbf{x}_t}\log p_t(\mathbf{x}_t)\right)\Delta t,$$

which match the Euler-Maruyama form of the backward diffusion SDE. Then, for $\boldsymbol{\mu}_t^{\text{FPS}}$ and $\Sigma_t^{\text{FPS}}$, we have:

$$\Sigma_t^{\text{FPS}} = \left((\Sigma_t^{\text{DDIM}})^{-1} + \frac{1}{\sigma^2 \cdot \overline{\alpha}_t} \cdot \boldsymbol{A}^\top \boldsymbol{A}\right)^{-1} = \beta(t)\Delta t \cdot \boldsymbol{I},$$

and

$$
\begin{aligned}
\boldsymbol{\mu}_t^{\text{FPS}} &= \Sigma_t^{\text{FPS}} \cdot \left(\left(\Sigma_t^{\text{DDIM}}\right)^{-1} \boldsymbol{\mu}_t^{\text{DDIM}}(\mathbf{x}_t) + \frac{1}{\sigma^2 \cdot \overline{\alpha}_t} \cdot \boldsymbol{A}^\top \mathbf{y}_t\right) \\
&= \left(\Sigma_t^{\text{DDIM}}(\Sigma_t^{\text{FPS}})^{-1}\right)^{-1} \cdot \boldsymbol{\mu}_t^{\text{DDIM}}(\mathbf{x}_t) + \frac{1}{\sigma^2 \cdot \overline{\alpha}_t}\Sigma_t^{\text{FPS}} \cdot (\boldsymbol{A}^\top \mathbf{y}_t) \\
&= \left(I - \frac{\beta(t)\Delta t}{\sigma^2 \cdot \overline{\alpha}_t}\boldsymbol{A}^\top \boldsymbol{A}\right) \cdot \boldsymbol{\mu}_t^{\text{DDIM}}(\mathbf{x}_t) + \frac{\beta(t)\Delta t}{\sigma^2 \cdot \overline{\alpha}_t} \cdot \boldsymbol{A}^\top \mathbf{y}_t \\
&= \mathbf{x}_t + \left(\frac{\beta(t)}{2}\mathbf{x}_t + \beta(t)\nabla_{\mathbf{x}_t}\log p_t(\mathbf{x}_t)\right)\Delta t + \frac{\beta(t)\Delta t}{\sigma^2 \cdot \overline{\alpha}_t} \cdot (\boldsymbol{A}^\top \mathbf{y}_t - \boldsymbol{A}^\top \boldsymbol{A}\mathbf{x}_t).
\end{aligned}
$$

We have already known $p_{\boldsymbol{\theta}}(\mathbf{y}_t \mid \mathbf{x}_t) \sim \mathcal{N}(\boldsymbol{A}\mathbf{x}_t, \sigma^2\overline{\alpha}_t \cdot I)$, it holds that:

$$\nabla_{\mathbf{x}_t}\log p(\mathbf{y}_t \mid \mathbf{x}_t) = \frac{1}{\sigma^2 \cdot \overline{\alpha}_t} \cdot (\boldsymbol{A}^\top \mathbf{y}_t - \boldsymbol{A}^\top \boldsymbol{A}\mathbf{x}_t).$$

Therefore, we can finally conclude that

$$
\begin{aligned}
\boldsymbol{\mu}_t^{\text{FPS}} &= \mathbf{x}_t + \left(\frac{\beta(t)}{2}\mathbf{x}_t + \beta(t)\nabla_{\mathbf{x}_t}\log p_t(\mathbf{x}_t)\right)\Delta t + \beta(t)\Delta t \cdot \nabla_{\mathbf{x}_t}\log p(\mathbf{y}_t \mid \mathbf{x}_t) \\
&= \mathbf{x}_t + \left(\frac{\beta(t)}{2}\mathbf{x}_t + \beta(t)\nabla_{\mathbf{x}_t}\log p_t(\mathbf{x}_t \mid \mathbf{y}_t)\right)\Delta t.
\end{aligned}
$$

Therefore, when time step $\Delta t \to 0$, the continuous limit of FPS should be

$$\mathrm{d}\mathbf{x}_t = \left[-\frac{\beta(t)}{2}\mathbf{x}_t - \beta(t)\nabla_{\mathbf{x}_t}\log p_t(\mathbf{x}_t \mid \mathbf{y}_t)\right]\mathrm{d}t + \sqrt{\beta(t)}\mathrm{d}\overline{\mathbf{W}}_t.$$

## D.2 THE CONSISTENCY OF FPS-SMC

In this section, we are going to use the method of induction to prove the Proposition 4.1. Before the official proof, I would like to introduce some useful properties on weak convergence over distributions. For simplicity of notations, we denote $p := p_{\boldsymbol{\theta}}$ in this proof.

**Lemma D.1.** *Assume a sequence of distributions over support set $\Omega$ satisfies $p_n \xrightarrow{w} p^*$ when $n \to \infty$. $c : \Omega \to \mathbb{R}^+$ is a continuous and bounded function over $\Omega$. Then: denote*

$$q_n(\mathbf{x}) = \frac{c(\mathbf{x})p_n(\mathbf{x})}{\int c(\mathbf{x})p_n(\mathbf{x})\mathrm{d}\mathbf{x}} \;\; and \;\; q^*(\mathbf{x}) = \frac{c(\mathbf{x})p^*(\mathbf{x})}{\int c(\mathbf{x})p^*(\mathbf{x})\mathrm{d}\mathbf{x}}.$$

*It holds that $q_n \xrightarrow{w} q^*$ when $n \to \infty$.*

**Lemma D.2.** *Assume a sequence of distributions over support set $\Omega$ satisfies $p_n \xrightarrow{w} p^*$ when $n \to \infty$. $q(\mathbf{x}' \mid \mathbf{x})$ is a conditional distribution. Denote*

$$r_n(\mathbf{x}') = \int q(\mathbf{x}' \mid \mathbf{x})p_n(\mathbf{x})\mathrm{d}\mathbf{x} \;\; and \;\; r^*(\mathbf{x}') = \int q(\mathbf{x}' \mid \mathbf{x})p^*(\mathbf{x})\mathrm{d}\mathbf{x}.$$

*Then it holds that $r_n \xrightarrow{w} r^*$ when $n \to \infty$.*

**Lemma D.3.** *Assume a sequence of distributions over support set $\Omega \times \Omega'$ satisfies $p_n(\mathbf{x}, \mathbf{x}') \xrightarrow{w} p^*(\mathbf{x}, \mathbf{x}')$ when $n \to \infty$. Denote*

$$q_n(\mathbf{x}) = \int p_n(\mathbf{x}, \mathbf{x}') \mathrm{d}\mathbf{x}' \quad and \quad q^*(\mathbf{x}) = \int p^*(\mathbf{x}, \mathbf{x}') \mathrm{d}\mathbf{x}'.$$

*Then it holds that $q_n \xrightarrow{w} q^*$ when $n \to \infty$.*

These three lemmas can be easily proved by the definition of weak convergence. According to Assumption 4.1, we know that $p(\mathbf{x}_k \mid \mathbf{x}_{k+1}) = p^*(\mathbf{x}_k \mid \mathbf{x}_{k+1})$ holds for all $k$. Since $p(\mathbf{y}_k \mid \mathbf{x}_k) = p^*(\mathbf{y}_k \mid \mathbf{x}_k)$ is fixed and known in both forward and backward processes, we have:

$$p(\mathbf{x}_k \mid \mathbf{x}_{k+1}, \mathbf{y}_k) \propto p(\mathbf{x}_k \mid \mathbf{x}_{k+1}) p(\mathbf{y}_k \mid \mathbf{x}_k) = p^*(\mathbf{x}_k \mid \mathbf{x}_{k+1}) p^*(\mathbf{y}_k \mid \mathbf{x}_k) \propto p^*(\mathbf{x}_k \mid \mathbf{x}_{k+1}, \mathbf{y}_k),$$

which leads to $p(\mathbf{x}_k \mid \mathbf{x}_{k+1}, \mathbf{y}_k) = p^*(\mathbf{x}_k \mid \mathbf{x}_{k+1}, \mathbf{y}_k)$. Now we start proving:

$$p(\mathbf{x}_k \mid \mathbf{y}_{k:N}) \xrightarrow{w} p^*(\mathbf{x}_k \mid \mathbf{y}_{k:N}) \quad \text{when } M \to \infty \tag{25}$$

by using the method of induction over $k$. We have known that $\{\mathbf{x}_N^{(j)}\}$ are $M$ i.i.d samples from $p^*(\mathbf{x}_N \mid \mathbf{y}_N)$, which means Eq. (25) holds for $k = N$. If the equation holds for $k + 1$, then $\{\mathbf{x}_{k+1}^{(j)}\}$ are i.i.d samples from $p(\mathbf{x}_{k+1} \mid \mathbf{y}_{k+1:N})$. For each $j \in [M]$, we generate

$$\overline{\mathbf{x}}_k^{(j)} \sim p(\mathbf{x}_k \mid \mathbf{x}_{k+1}^{(j)}, \mathbf{y}_k).$$

For each $j$, $(\overline{\mathbf{x}}_k^{(j)}, \mathbf{x}_{k+1}^{(j)})$ follows the distribution

$$q(\overline{\mathbf{x}}_k, \mathbf{x}_{k+1} \mid \mathbf{y}_{k:N}) := p(\overline{\mathbf{x}}_k \mid \mathbf{x}_{k+1}, \mathbf{y}_k) \cdot p(\mathbf{x}_{k+1} \mid \mathbf{y}_{k+1:N}) \xrightarrow{w} p^*(\overline{\mathbf{x}}_k \mid \mathbf{x}_{k+1}, \mathbf{y}_k) \cdot p^*(\mathbf{x}_{k+1} \mid \mathbf{y}_{k+1:N}).$$

The next step is the resampling, whose weights are $c(\overline{\mathbf{x}}_k^{(j)}, \mathbf{x}_{k+1}^{(j)}) = p(\mathbf{y}_k \mid \mathbf{x}_{k+1}^{(j)})$. Since $p(\mathbf{y}_k \mid \mathbf{x}_{k+1})$ is a Gaussian distribution, the weight function $c(\cdot, \cdot)$ is continuous and bounded. For each continuous bounded function $f$ and each $j \in [M]$, by the Law of Large Numbers:

$$\begin{aligned}
\mathbb{E}f(\mathbf{x}_k, \mathbf{x}_{k+1}) &= \frac{\sum_{j=1}^M c(\overline{\mathbf{x}}_k^{(j)}, \mathbf{x}_{k+1}^{(j)}) \cdot f(\overline{\mathbf{x}}_k^{(j)}, \mathbf{x}_{k+1}^{(j)})}{\sum_{j=1}^M c(\overline{\mathbf{x}}_k^{(j)}, \mathbf{x}_{k+1}^{(j)})} \\
&\xrightarrow{a.s.} \frac{\int c(\mathbf{x}_k, \mathbf{x}_{k+1}) f(\mathbf{x}_k, \mathbf{x}_{k+1}) \cdot q(\mathbf{x}_k, \mathbf{x}_{k+1} \mid \mathbf{y}_{k:N}) \mathrm{d}\mathbf{x}_k \mathrm{d}\mathbf{x}_{k+1}}{\int c(\mathbf{x}_k, \mathbf{x}_{k+1}) \cdot q(\mathbf{x}_k, \mathbf{x}_{k+1} \mid \mathbf{y}_{k:N}) \mathrm{d}\mathbf{x}_k \mathrm{d}\mathbf{x}_{k+1}} \\
&= \int f(\mathbf{x}_k, \mathbf{x}_{k+1}) r(\mathbf{x}_k, \mathbf{x}_{k+1} \mid \mathbf{y}_{k:N}) \mathrm{d}\mathbf{x}_k \mathrm{d}\mathbf{x}_{k+1}
\end{aligned}$$

which implies $p(\mathbf{x}_k, \mathbf{x}_{k+1} \mid \mathbf{y}_{k:N}) \xrightarrow{w} r(\mathbf{x}_k, \mathbf{x}_{k+1} \mid \mathbf{y}_{k:N})$ where

$$r(\mathbf{x}_k, \mathbf{x}_{k+1} \mid \mathbf{y}_{k:N}) \propto c(\mathbf{x}_k, \mathbf{x}_{k+1}) \cdot q(\mathbf{x}_k, \mathbf{x}_{k+1} \mid \mathbf{y}_{k:N}).$$

By using Lemma D.1,

$$\begin{aligned}
r(\mathbf{x}_k, \mathbf{x}_{k+1} \mid \mathbf{y}_{k:N}) \xrightarrow{w} r^*(\mathbf{x}_k, \mathbf{x}_{k+1} \mid \mathbf{y}_{k:N}) &\propto c(\mathbf{x}_k, \mathbf{x}_{k+1}) p(\mathbf{x}_k \mid \mathbf{x}_{k+1}, \mathbf{y}_k) \cdot p^*(\mathbf{x}_{k+1} \mid \mathbf{y}_{k+1:N}) \\
&= p(\mathbf{y}_k \mid \mathbf{x}_{k+1}) p(\mathbf{x}_k \mid \mathbf{x}_{k+1}, \mathbf{y}_k) \cdot p^*(\mathbf{x}_{k+1} \mid \mathbf{y}_{k+1:N}) \propto p^*(\mathbf{x}_k, \mathbf{x}_{k+1} \mid \mathbf{y}_{k:N}),
\end{aligned}$$

and it finally leads to

$$p(\mathbf{x}_k, \mathbf{x}_{k+1} \mid \mathbf{y}_{k:N}) \xrightarrow{w} p^*(\mathbf{x}_k, \mathbf{x}_{k+1} \mid \mathbf{y}_{k:N}).$$

By Lemma D.3, it holds that:

$$p(\mathbf{x}_k \mid \mathbf{y}_{k:N}) \xrightarrow{w} p^*(\mathbf{x}_k \mid \mathbf{y}_{k:N}).$$

Now we finish the induction. Let $k = 0$, we have:

$$p(\mathbf{x}_0 \mid \mathbf{y}_{0:N}) \xrightarrow{w} p^*(\mathbf{x}_0 \mid \mathbf{y}_{0:N}).$$

After taking expectation over $p(\mathbf{y}_{1:N} \mid \mathbf{y}_0)$ and using Lemma D.2:

$$p(\mathbf{x}_0 \mid \mathbf{y}_0) \xrightarrow{w} p^*(\mathbf{x}_0 \mid \mathbf{y}_0),$$

which comes to our conclusion.

# E  ADDITIONAL EXPERIMENTAL RESULTS

We first outline the configurations for the five tasks we cast as linear inverse problems in image processing. (a) Inpainting task with random-type: we mask each pixel in the image at a 92% probability, affecting all three RGB channels. (b) Inpainting task with box-type: we select a random $128 \times 128$ box-shaped region within the $256 \times 256$ image and mask out all the pixels within this box, across all three RGB channels. (c) Super resolution: we downscale our $256 \times 256$ images to $64 \times 64$ by employing bicubic downsampling with a factor of 4. (d) Gaussian deblurring and motion deblurring: The shape of the Gaussian kernel is $61 \times 61$ with its intensity to be 3.0, while for motion deblurring, we apply a kernel of the same size, but with an intensity of 0.5. As we know, the kernel of Gaussian deblurring is separable, which faciliates memory-efficient computation of linear mappings in the form of $(\lambda \boldsymbol{I} + \boldsymbol{A}^\top \boldsymbol{A})^{-1}$. Following Kawar et al. (2022), we replace traditional motion deblurring with anisotropic Gaussian deblurring in our experiments, mirroring the approach used in DDRM.

Next, we present our additional experimental outcomes on the FFHQ $256 \times 256$-1k validation dataset and ImageNet $256 \times 256$-1k validation dataset on the other two metrics, PSNR and SSIM, in Tables 3 and 4. The results show that our proposed methods achieve higher PSNR values compared to baseline approaches. However, they do not surpass the best existing methods in terms of the SSIM metric. Overall, our FPS and FPS-SMC methods exceed the performance of all current models on the majority of relevant tasks when evaluated using PSNR, LPIPS and FID metrics. Moreover, we illustrate in 6 - 9 the generated figures by our two methods as well as the corresponding inputs and measurements for inpainting (box), super resolution, Gaussian deblurring and inpainting (random) tasks on both FFHQ and ImageNet datasets.

Observing the performance in the inpainting (random) task where 92% of all the pixels are masked out, our methods successfully reconstruct most of the labels, which demonstrates their robustness in handling high undersampling factors. Meanwhile, in the inpainting (box) task, while the generated samples do not perfectly align with the labels, which is a common issue in this task due to the absence of information in the masked image area, they remain structurally consistent and semantically cohesive with the unmasked portions. Moreover, with the FFHQ dataset, our model's capability to generate human faces with varied facial expressions further validates its strength in preventing mode collapse and maintaining diversity.

Table 3: Quantitative results (PSNR, SSIM) of our model and existing models on a various of linear inverse problems on FFHQ $256 \times 256$-1k validation dataset.

| FFHQ | Super Resolution | | Inpainting (box) | | Gaussian Deblur | | Inpainting (random) | | Motion Deblur | |
|---|---|---|---|---|---|---|---|---|---|---|
| Methods | PSNR | SSIM | PSNR | SSIM | PSNR | SSIM | PSNR | SSIM | PSNR | SSIM |
| **FPS** | 27.48 | 0.807 | 24.17 | 0.865 | 26.45 | 0.773 | 26.79 | 0.820 | 26.70 | 0.828 |
| **FPS-SMC** | **28.10** | 0.807 | **24.70** | 0.862 | **26.54** | 0.773 | **27.33** | 0.819 | **27.39** | 0.826 |
| DPS | 25.67 | 0.852 | 22.47 | **0.873** | 24.25 | 0.811 | 25.23 | **0.851** | 24.92 | **0.859** |
| DDRM | 25.36 | 0.835 | 22.24 | 0.869 | 23.36 | 0.767 | 9.19 | 0.319 | - | - |
| MCG | 20.05 | 0.559 | 19.97 | 0.703 | 6.72 | 0.051 | 21.57 | 0.751 | - | - |
| PnP-ADMM | 26.55 | **0.865** | 11.65 | 0.642 | 24.93 | **0.812** | 8.41 | 0.325 | - | - |
| Score-SDE | 17.62 | 0.617 | 18.51 | 0.678 | 7.12 | 0.109 | 13.52 | 0.437 | - | - |
| ADMM-TV | 23.86 | 0.803 | 17.81 | 0.814 | 22.37 | 0.801 | 22.03 | 0.784 | - | - |

# F  ABLATION STUDIES

## F.1  FORWARD AND BACKWARD GENERATION OF $y$-SEQUENCE

In this section, we focus on the $\{\mathbf{y}_k\}_{k=0}^N$-sequence employed in the filtering-based methods we propose. To synchronize the $\{\mathbf{x}_k\}_{k=0}^N$-sequence with the $\{\mathbf{y}_k\}_{k=0}^N$-sequence, we construct our

Table 4: Quantitative results (PSNR, SSIM) of our model and existing models on a various of linear inverse problems on ImageNet $256 \times 256$-1k validation dataset.

| Methods | Super Resolution | | Inpainting (box) | | Gaussian Deblur | | Inpainting (random) | | Motion Deblur | |
|---|---|---|---|---|---|---|---|---|---|---|
| | PSNR | SSIM | PSNR | SSIM | PSNR | SSIM | PSNR | SSIM | PSNR | SSIM |
| **FPS** | 24.32 | 0.724 | 20.16 | 0.752 | 23.58 | 0.581 | 23.39 | 0.688 | 22.71 | 0.598 |
| **FPS-SMC** | 24.78 | 0.731 | **22.03** | 0.748 | **23.81** | 0.599 | **24.12** | 0.685 | **23.27** | 0.614 |
| DPS | 23.87 | 0.781 | 18.90 | 0.794 | 21.97 | **0.706** | 22.20 | **0.739** | 20.55 | **0.634** |
| DDRM | **24.96** | **0.790** | 18.66 | **0.814** | 22.73 | 0.705 | 14.29 | 0.403 | - | - |
| MCG | 13.39 | 0.227 | 17.36 | 0.633 | 16.32 | 0.441 | 19.03 | 0.546 | - | - |
| PnP-ADMM | 23.75 | 0.761 | 12.70 | 0.657 | 21.81 | 0.669 | 8.39 | 0.300 | - | - |
| Score-SDE | 12.25 | 0.256 | 16.48 | 0.612 | 15.97 | 0.436 | 18.62 | 0.517 | - | - |
| ADMM-TV | 22.17 | 0.679 | 17.96 | 0.785 | 19.99 | 0.634 | 20.96 | 0.676 | - | - |

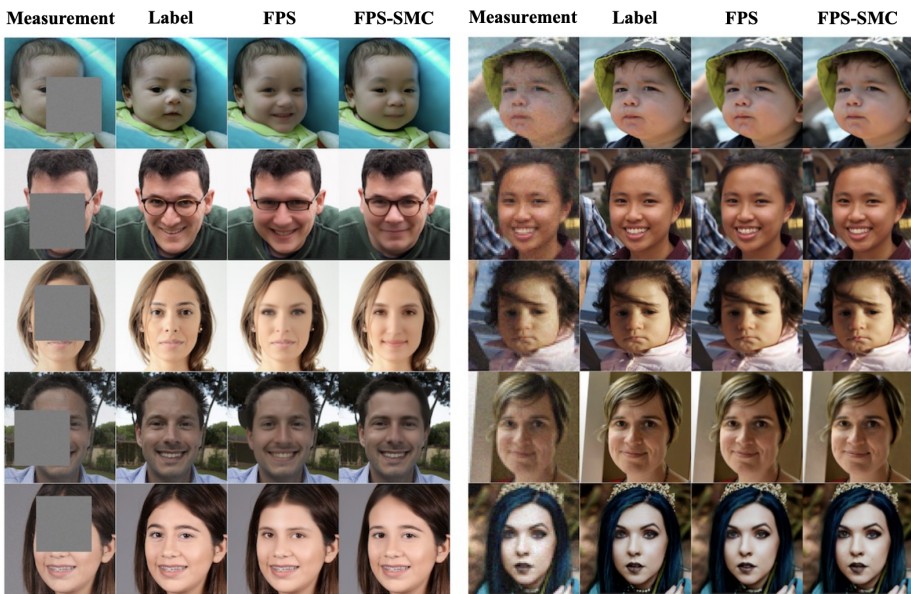

Figure 6: Examples for inpainting (box) and super resolution over FFHQ dataset

$\{\mathbf{y}_k\}_{k=0}^N$-sequence by using a DDIM backward process framework, with an explicit unconditional score function. Unlike existing posterior sampling methods such as DPS (Chung et al., 2022a) where $\mathbf{y}_1 = \mathbf{y}_2 = \ldots = \mathbf{y}_N = \mathbf{y}$ or conditional score-based generative models (Song et al., 2020b) that sample $\mathbf{y}_k \sim \mathcal{N}(\mathbf{y}, \sigma_k^2 \boldsymbol{I})$, the noise introduced to $\mathbf{y}$ at each step in our methods is correlated as shown in Eq. (26)

$$\mathbf{y}_k = \sqrt{1 - \beta_k} \cdot \mathbf{y}_{k-1} + \sqrt{\beta_k} \cdot \boldsymbol{A}\mathbf{z}_k, \qquad \mathbf{z}_k \sim \mathcal{N}(0, \boldsymbol{I}) \text{ are independent with each other}, \quad (26)$$

It ensures that our generated sequence $\{\mathbf{y}_k\}_{k=0}^N$ is continuous, with incremental noise added at each step.

Here, we conduct separate experiments to demonstrate the contrast between these two different ways of generating the $\{\mathbf{y}_k\}_{k=0}^N$ sequence. In the FPS method proposed by us, we use DDIM backward process to make Eq. (26) holds. Conversely, in the control group, the FPS with Independent Noise (FPS-IN) model generates the $\{\mathbf{y}_k\}_{k=0}^N$ sequence in a simpler manner by:

$$\mathbf{y}_k = \sqrt{\overline{\alpha}_k}\mathbf{y} + \sqrt{1 - \overline{\alpha}_k} \cdot \boldsymbol{A}\mathbf{z}_k, \qquad \mathbf{z}_k \sim \mathcal{N}(0, \boldsymbol{I}) \text{ are independent with each other}.$$

Meanwhile, to ensure the validity of our ablation study, we maintain all other experimental conditions identical. From the following Table 5, we can see that in all the five tasks and both of the two datasets,

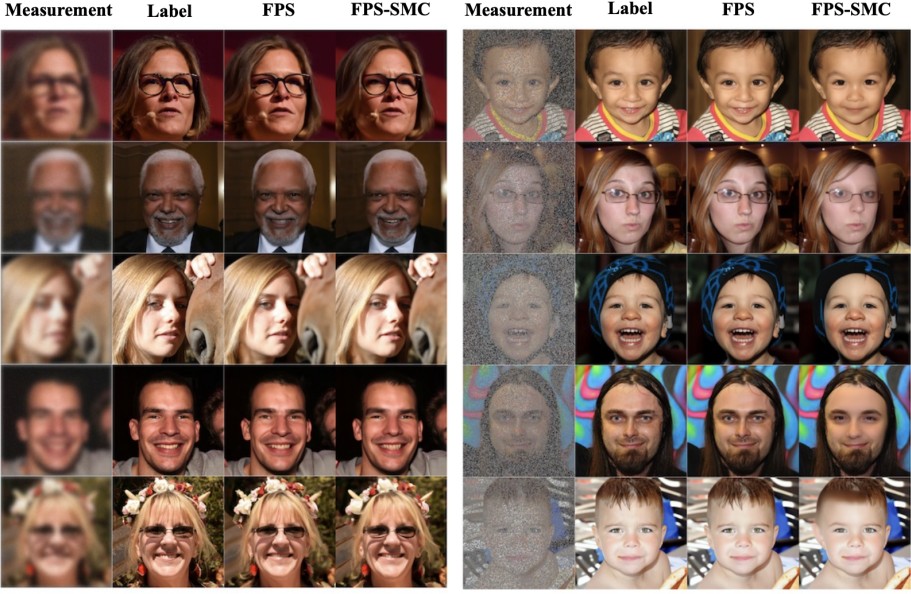

Figure 7: Examples for Gaussian deblurring and inpainting (random) over FFHQ dataset

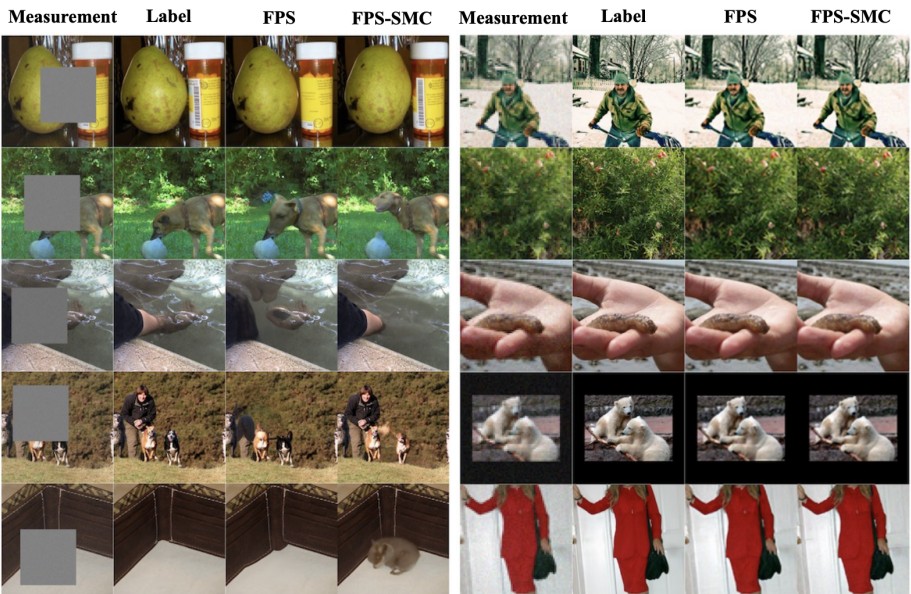

Figure 8: Examples for inpainting (box) and super resolution over ImageNet dataset

FPS consistently outperforms the control group, FPS-IN. This outcome affirms the significance of the additive noise correlation in the $\{\mathbf{y}_k\}_{k=0}^N$ sequence for the model's performance.

Furthermore, we generate two sets of images by using FPS and the control group, FPS-IN, for the inpainting (box) task, with the outcomes presented in Fig. 10. Both methods of generating $\{\mathbf{y}_k\}_{k=0}^N$ sequence enable our model to perform effectively, creating well-composed images. Comparatively, the images generated by FPS tend to be more accurate and closely match the unmasked portions of the labels.

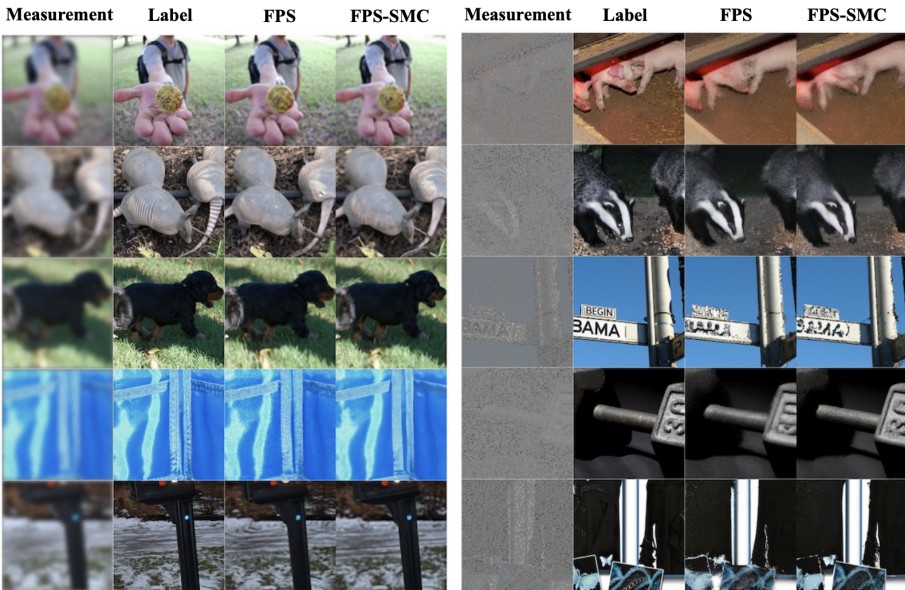

| Measurement | Label | FPS | FPS-SMC | Measurement | Label | FPS | FPS-SMC |

Figure 9: Examples for Gaussian deblurring and inpainting (random) over ImageNet dataset

Table 5: Quantitative results (FID, LPIPS) of FPS and our control group, FPS-IN on FFHQ $256 \times 256$-1k validation dataset and ImageNet $256 \times 256$-1k validation dataset.

| FFHQ | Super Resolution | | Inpainting (box) | | Gaussian Deblur | | Inpainting (random) | | Motion Deblur | |
|---|---|---|---|---|---|---|---|---|---|---|
| **Methods** | FID | LPIPS | FID | LPIPS | FID | LPIPS | FID | LPIPS | FID | LPIPS |
| **FPS** | **26.66** | **0.212** | **26.13** | **0.141** | **30.03** | **0.248** | **35.21** | **0.265** | **26.18** | **0.221** |
| FPS-IN | 27.18 | 0.247 | 27.84 | 0.169 | 32.79 | 0.251 | 39.90 | 0.316 | 26.86 | 0.239 |
| ImageNet | Super Resolution | | Inpainting (box) | | Gaussian Deblur | | Inpainting (random) | | Motion Deblur | |
| Methods | FID | LPIPS | FID | LPIPS | FID | LPIPS | FID | LPIPS | FID | LPIPS |
| **FPS** | **47.32** | **0.329** | **33.19** | **0.204** | **54.41** | **0.396** | **42.68** | **0.325** | **52.22** | **0.370** |
| FPS-IN | 50.06 | 0.541 | 34.21 | 0.255 | 56.21 | 0.409 | 44.40 | 0.337 | 53.10 | 0.389 |

## F.2 RUNNING TIME OF FPS-SMC WITH INCREASING PARTICLE SIZE

In this section, we compare the wall-clock running time of FPS and FPS-SMC with other popular posterior sampling models. By using a single A100 GPU, the running time for each algorithm is listed in Table 6. As we can see, although not the fastest, FPS provides running speed faster than average for the posterior sampling in the inpainting (box) task of FFHQ dataset. Furthermore, we study the running time of FPS-SMC with increasing particle size. Despite the fact that the computational cost should be proportional to the particle size $M$, we can reduce the running time by batch-wise operators while coding. According to Table 7, the running time $t(M)$ approximately holds $t(M) \propto \sqrt{M}$.

## F.3 INFLUENCE OF THE NOISE LEVEL IN DDIM FRAMEWORK

Within the DDIM unconditional generative framework, the noise level $c \in [0,1]$ serves as a pre-determined hyper-parameter. According to our experiments, we notice that varying $c$ leads to distinct generated images, with each task presenting an optimal choice of $c$. Therefore, in this section, we explore how the noise level $c$ affects the quality of generated images. As evident in Figs. 11 and 12, an excessively high $c$ value results in over-smoothed images, while a very low $c$ causes insufficient smoothing, degrading image quality. We thus select task-specific $c$ values, with our chosen parameters detailed in Table 8.

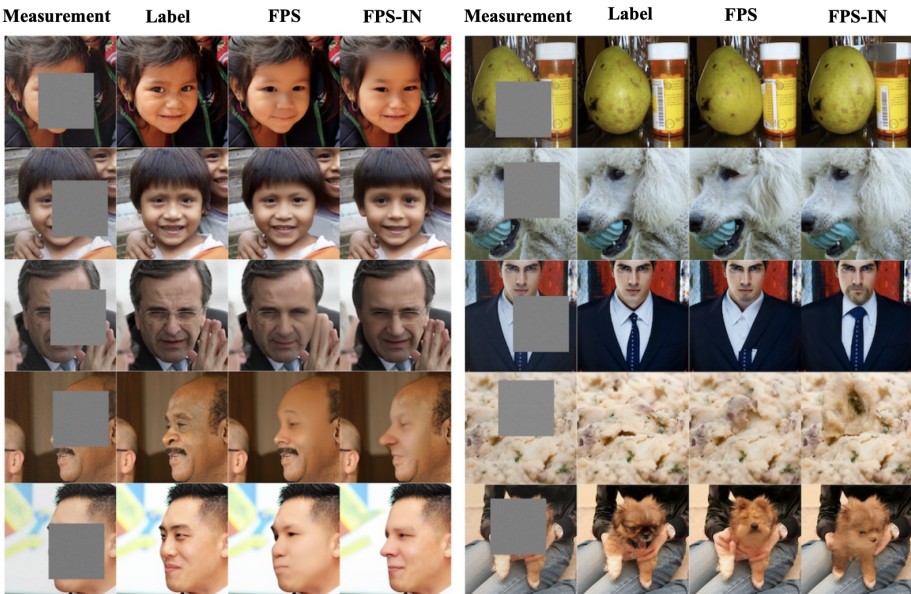

Figure 10: Comparison of FPS and the control group, FPS-IN under the inpainting (box) task on FFHQ and ImageNet dataset

Table 6: Wall-clock running time for a single posterior sampling on the inpainting (box) task of FFHQ-1k-validation dataset, by using a single A100 GPU

| Model | Running time (in seconds) |
|---|---|
| **FPS** (our paper) | 33.07 |
| Score-SDE (Song et al., 2020b) | 32.93 |
| DPS (Chung et al., 2022a) | 70.42 |
| DDRM (Kawar et al., 2022) | 2.034 |
| ΠGDM (Song et al., 2023a) | 33.18 |
| MCG (Chung et al., 2022b) | 73.16 |
| PnP-ADMM (Chan et al., 2016) | 3.595 |

Table 7: The connection between the running time of FPS-SMC and the particle size $M$, by using a single A100 GPU

| Particle Size | Running time (in seconds) |
|---|---|
| $M = 1$ (FPS) | 33.07 |
| $M = 2$ | 39.15 |
| $M = 5$ | 57.88 |
| $M = 10$ | 82.12 |
| $M = 20$ | 116.90 |
| $M = 100$ | 283.53 |

Table 8: Different $c$ values we choose in various tasks over datasets FFHQ and ImageNet

| Dataset | Super Resolution | Inpainting (box) | Gaussian Deblur | Inpainting (random) | Motion Deblur |
|---|---|---|---|---|---|
| FFHQ | 0.3 | 0.95 | 0.3 | 0.95 | 0.3 |
| ImageNet | 0.15 | 0.25 | 0.3 | 0.25 | 0.3 |

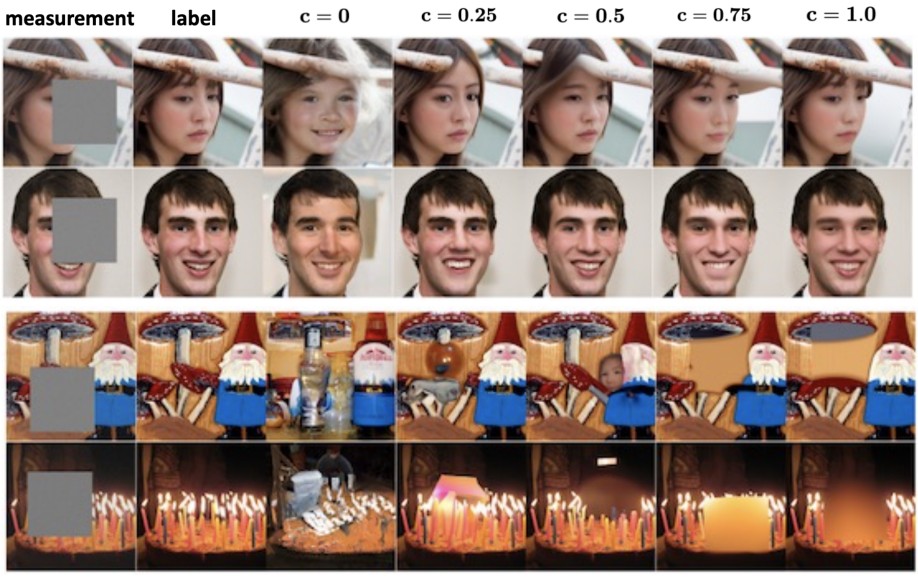

Figure 11: Generated FFHQ images by FPS with different noise level in inpainting (box) task

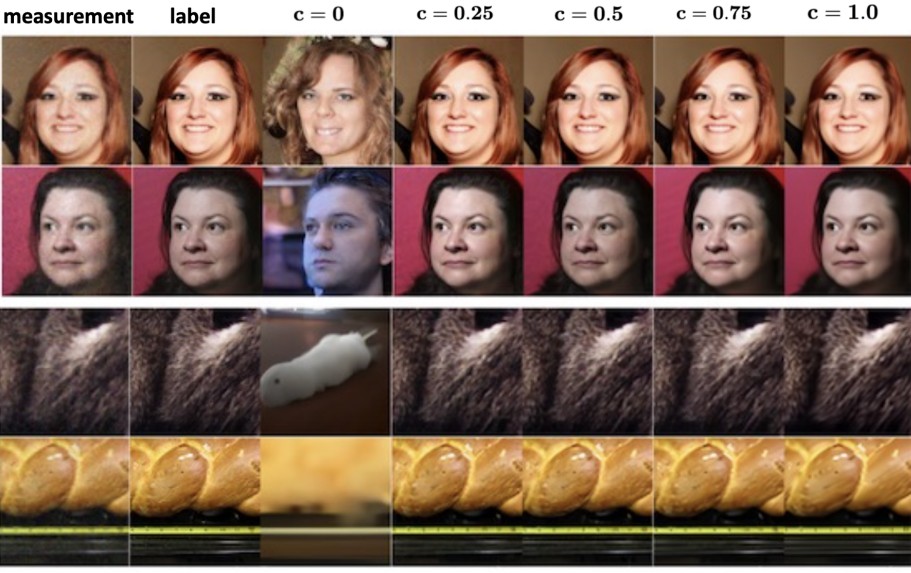

Figure 12: Generated ImageNet images by FPS with different noise level in super resolution task

