# OpenReview forum: "Diffusion Posterior Sampling for Linear Inverse Problem Solving: A Filtering Perspective"
_ICLR.cc/2024/Conference — ICLR 2024 poster_

### Official Review · Reviewer_cbzu · 2023-10-28

**Soundness:** 4 excellent
**Presentation:** 4 excellent
**Contribution:** 4 excellent
**Rating:** 10
**Confidence:** 4

**Summary:**

This paper provides a link between Bayesian posterior sampling and Bayesian filtering in diffusion models for linear inverse problems. In general, exact posterior sampling is intractable. To overcome this difficulty, this paper introduces a diffusion process to the measurement vector and incorporates the information of the measurement by Bayesian filtering methods in the backward process.  In the ideal case, the exactness of the proposed method is theoretically guaranteed, which is realized by the sequential Monte Carlo method using infinitely many particles. Numerical experiments show that the performance of the method is as good as those of the SOTA methods.

**Strengths:**

This article provides an important perspective that has been missing until now for the linear inverse problem using diffusion models. The proposed method is reasonable in the Bayesian framework, and achieves as good performance as the current SOTA methods.

**Weaknesses:**

Extendability to nonlinear measurement cases is not clear. FPS-SMC, which uses multiple particles, requires considerably heavy computational cost.

**Questions:**

It is possible to extend the proposed method for nonlinear measurement cases?

---

> ### Author Response · Authors · 2023-11-23
>
> We really appreciate your positive feedback. Please find our response to your questions in 'general questions proposed by reviewers'.

---

### Official Review · Reviewer_pD39 · 2023-11-01

**Soundness:** 3 good
**Presentation:** 3 good
**Contribution:** 3 good
**Rating:** 6
**Confidence:** 4

**Summary:**

This paper proposed a filtering perspective on the linear inverse problem with diffusion models, which yields a novel FPS algorithm. Moreover, this paper gave a theoretical proof that that the FPS algorithm correctly samples from the Bayesian posterior
distribution as the number of particles approaches infinity.

**Strengths:**

1. A novel filtering perspective on solving linear inverse problems using diffusion models, which is interesting and inspiring.
2. A theoretical proof is given that the FPS algorithm can correctly sample from the Bayesian posterior
distribution as the number of particles approaches infinity.

**Weaknesses:**

1. Compared to previous methods like DPS (in fact, currently, DPS is no longer the SOTA), the improvement seems marginal in some cases, and even worse in some cases like inpainting. From my understanding, inpainting is among the most challenging tasks for linear image restoration tasks and the inferiority of FPS might suggest some underlying unaddressed problem for this proposed method.

2. There is a lack of complexity or running time analysis. How fast is FPS compared with other methods? How does the time increase with a different number of particles M?

**Questions:**

1. How did you add noise with standard deviation \sigma= 0.05? Did you account for the scaling of [-1, 1] so that the actual standard deviation added is 2 * \sigma=  0.05*2 = 1, as did in DDRM? The code is not provided so that I cannot check this point.

---

> ### Author Response · Authors · 2023-11-23
>
> Thank you for taking the time to review our work. Please find our response to your questions on running time analysis and the code release in 'general questions proposed by reviewers'. We address the other questions below.
>
> Q1: Compared to previous methods like DPS (in fact, currently, DPS is no longer the SOTA), the improvement seems marginal in some cases, and even worse in some cases like inpainting. From my understanding, inpainting is among the most challenging tasks for linear image restoration tasks and the inferiority of FPS might suggest some underlying unaddressed problem for this proposed method.
>
> A: We agree with the reviewer that the improvement is marginal in some cases, but would like to argue that the improvement is significant for most tasks we consider. Note that even for the inpainting task, our methods perform better on FFHQ (random inpainting) in PSNR, which is the classical metric in inverse problem solving. We believe the value of our approach is not just empirical performance, but also its strong theoretical guarantee, which is the first of its kind.
>
> Q2: How did you add noise with standard deviation $\sigma= 0.05$? Did you account for the scaling of $[-1, 1]$ so that the actual standard deviation added is $2 * \sigma= 0.05*2 = 0.1$, as did in DDRM?
>
> A: Yes, we follow DDRM to process our images and add noise. We will release our code upon publication.

---

### Official Review · Reviewer_x48M · 2023-11-01

**Soundness:** 3 good
**Presentation:** 3 good
**Contribution:** 3 good
**Rating:** 6
**Confidence:** 4

**Summary:**

This work introduces an efficient, asymptotically accurate diffusion sampling algorithm for linear inverse problems, linking Bayesian posterior sampling to Bayesian filtering in diffusion models, using sequential Monte Carlo methods for filtering, requiring no model re-training, and outperforming or matching existing methods in tasks like image inpainting, super-resolution, and motion deblur.

**Strengths:**

Solving inverse problems is a fundamental problem in machine learning and other fields. Diffusion models have a good potential to solve those problems, and it is a good attempts to provide a solution based on the filtering perspective.

**Weaknesses:**

The literature review for diffusion-based inverse problems are quite limited. It would be very important to compare with previous work to demonstrate the advantage and necessity of the new algorithm.

The scale of empirical experiments are relatively small.

**Questions:**

Is there a running time or NFE evaluation for the algorithm?

Although the algorithm is designed to solve linear inverse problems, does it work on non-linear ones, which may happen in real practice? For DPS, they have shown the possibilities to solve nonlinear problems.

---

> ### Author Response · Authors · 2023-11-23
>
> Thank you for taking the time to review our work. Please find our response to your questions in 'general questions proposed by reviewers'. In addition, we address the following question.
>
> Q1: The scale of our empirical experiments is relatively small.
>
> A: We have limited access to a single A100 GPU which strongly restricts the scale of our experiments. That said, we have carried out experiments to solve five different linear inverse problems for both FFHQ and ImageNet, which matches the setup in many existing papers on the same topic.

---

### Official Review · Reviewer_b9JP · 2023-11-04

**Soundness:** 1 poor
**Presentation:** 3 good
**Contribution:** 2 fair
**Rating:** 3
**Confidence:** 5

**Summary:**

This paper presents an SMC algorithm for sampling from the posterior of an inverse problem where the prior distribution distribution is a diffusion model, which is a natural way of sampling from the posterior of the diffusion model, due to the sequential structure of the sampling procedure. The resulting procedure does not require any additional training and, as claimed by the authors, samples from the posterior in the limit of infinite particles.

**Strengths:**

- The proposed method provides impressive results on inpainting tasks. Interestingly, other methods for inpainting with a diffusion prior often suffer from inconsistencies at the the borders of the patch. This is not the case of this method.

**Weaknesses:**

- I believe that method proposed in the paper is technically flawed. In the section 3 of the main paper, the authors give the state space model that they consider where notably, the use the same noise at step $k$ for both the observation and the state. They then proceed to claim that it holds that $p(x_0 | y_0) = \int p(x_0 | y_{0:N}) p(y_{1:N} | y_0) \mathrm{d}y_{1:N}$ and thus, to sample from $p(x_0 | y_0)$ it is enough to sample $p(y_{1:N} | y_0)$ and then sample from $p(x_0 | y_{0:n})$ using an SMC algorithm. However, this is not correct since $p(x_0 | y_{0:N}) = p(x_0 | y_0)$ for the state model that they consider. To show that this is the case, consider the case $N=1$. Then, the joint distribution of the SSM they consider is the following, since they share the noise:
$$
p(y_{0:1}, x_{0:1}) = p(y_0 | x_0) p(x_0) \int p(\mathrm{d} z) \delta_{a_1 x_0 + b_1 z} (x_1)   \delta_{a_1 y_0 + b_1 A z} (y_1)
$$
and thus,
$$
p(x_0 | y_{0:1}) = \frac{\int p(y_{0:1}, x_{0:1}) \mathrm{d} x_{1}}{\int p(y_{0:1}, x_{0:1}) \mathrm{d} x_{1} \mathrm{d} x_0} = \frac{p(y_0 | x_0) p(x_0) p(y_1 | y_0)}{\int p(y_0 | x_0) p(x_0) p(y_1 | y_0) \mathrm{d} x_0} = p(x_0 | y_0)
$$
Besides this fact, the methodology developed in the rest of the paper does not in fact sample from the correct posterior asymptotically. To see why this is the case, note that the state space model on which the authors apply SMC is the following:
$$
p_\theta(x_{0:N}, y_{0:N}) = p_N(y_N | x_N) p_N(x_N) \prod_{s = 0}^{N-1} p_\theta(x_s | x_{s+1}) p(y_s | x_s)
$$
It is straightforward to see that the p_\theta(x_0 | y_0) resulting from this model is **not** the target posterior $p^*(x_0 | y_0) \propto p(y_0 | x_0) p_\theta(x_0)$.

- The idea of using the same noise for the forward process and the observations is not new and is used in [1], which the authors cite but they fail to mention that they use the same idea. The authors also fail to mention author work on SMC applied to diffusion posterior sampling [2]. In fact in this paper the authors use a specific decomposition of the posterior, similar to what is claimed in the text box at the end of the section 3 of this paper. In contrast, their decomposition is theoretically justified but holds under stringent assumptions on the backward process, further confirming that what is claimed in this paper is not true.

- Finally, this paper is not the first one to develop a consistent diffusion posterior sampling algorithm, see [3] and [4] which develop **principled** SMC algorithms for asympotitcally exact posterior sampling. This is not a criticism, as these papers have been released 3/4 months ago and I understand that the authors may not have had knowledge of them.


[1] Song, Yang, et al. "Solving inverse problems in medical imaging with score-based generative models." arXiv preprint arXiv:2111.08005 (2021).

[2] Trippe, Brian L., et al. "Diffusion probabilistic modeling of protein backbones in 3d for the motif-scaffolding problem." arXiv preprint arXiv:2206.04119 (2022).

[3] Wu, Luhuan, et al. "Practical and asymptotically exact conditional sampling in diffusion models." arXiv preprint arXiv:2306.17775 (2023).

[4] Cardoso, Gabriel, et al. "Monte Carlo guided Diffusion for Bayesian linear inverse problems." arXiv preprint arXiv:2308.07983 (2023).

**Questions:**

i have no further questions

---

> ### Author Response · Authors · 2023-11-23
>
> Thank you so much for your thoughtful feedback. For literature review, please find our response in `general questions proposed by reviewers'. For the other question, we answer it below.
>
> Q1: The method proposed in this paper is technically flawed.
>
> A: There are in fact two state space models (SSM) in our method. The first SSM is given by the forward perturbation process where $\{x_k\}$ and $\{y_k\}$ share the same noise in each time step, so that $q(y_k \mid x_k) = \mathcal N(0, c_k^2 \sigma^2  I)$ holds for each $k$. In this forward graphical model, the following equation as pointed out by the reviewer is correct:
>
> $$q(x_0\mid y_{0:N}) = q(x_0 \mid y_0).$$
>
> In fact, this equation is correct even without 'noise sharing', because $x_0$ and $(y_k),k\geq 1$ are conditionally independent given $y_0$. We leverage this forward SSM to generate the $(y_k), 0\leq k\leq n$ sequence, after which we ignore the generated $(x_k),0\leq k\leq N$ and start the backward Bayesian filtering process based on $(y_k),0\leq k\leq N$. This assumes the forward and backward SSMs define the same joint distribution, which is valid when the diffusion model is optimal and no discretization errors exist.
>
> The second SSM is derived from the backward diffusion process. As the reviewer correctly noticed, it takes the form below:
>
> $$p_\theta(x_{0:N}, y_{0:N}) = p_N(x_N)p_N(y_N\mid x_N)\cdot \prod_{k=0}^{N-1} p_\theta(x_k \mid x_{k+1}) p(y_k\mid x_k)$$
>
> where $p_{\theta}(x_k \mid x_{k+1})$ can be obtained by running the SDE solver for one step. By using Bayesian filtering and Sequential Monte Carlo (SMC), we sample particles $x_k \sim p_\theta(x_k \mid y_{k:N})$ for $k=N,N-1,\ldots, 0$. Proposition 4.1 in our paper shows that, when we have perfect score estimator and accurate backward SDE solvers (without discretization error), these two SSMs will be equivalent. When particle size $M\rightarrow\infty$, we sample from the correct posterior distribution $p_\theta(x_0\mid y_{0:N}) \rightarrow q(x_0 \mid y_{0:N})$. Here, we denote $q$ as the distribution of the forward process and $p_\theta$ as the distribution of the backward process (SSM), just like the notations in our revision. Note the assumptions we required here are very similar to [3], which means that the contradiction proposed by the reviewer is not true.
>
> [3] Brian L Trippe, Jason Yim, Doug Tischer, David Baker, Tamara Broderick, Regina Barzilay, and Tommi Jaakkola. Diffusion probabilistic modeling of protein backbones in 3d for the motif-scaffolding problem. arXiv preprint arXiv:2206.04119, 2022.

---

> > ### Comment · Reviewer_b9JP · 2023-11-23
> >
> > Thank your for your response. When I posted my initial review there was no mention of this assumption in the main paper. I believe that this assumption, which is far from true in practice,  is not needed to construct a consistent sampling procedure, as is done in [3] and [4]. I maintain my score.

---

### Author Response · Authors · 2023-11-23
**General Questions Proposed by Reviewers**

We thank all reviewers for their valuable feedback. We have updated our paper with improvements summarized below:
1. We improved the overall writing quality of our paper with better motivation, exposition, and fixing several typos in the draft.
2. We included a discussion with several additional papers in the introduction, background, and additionally provided one related work paragraph in appendix.
3. We summarized all assumptions for Proposition 4.1 in Assumption 4.1. Unlike before, we clearly introduce these assumptions and compare them with other relevant papers.
4. In Section 5.3, ``Ablation study: Influence of the particle size", we added an analysis on the relationship between the running time of FPS-SMC and the particle size $M$. We also made a table showing that the wall-clock time for running FPS-SMC is approximately $t\propto \sqrt{M}$. We provide additional experimental details in Appendix F.2.

Below we address concerns shared by multiple reviewers.

Q1: Generalization to nonlinear measurements

A: The focus of this work is on linear inverse problems. We would like to emphasize that linear inverse problems are important in their own right. As demonstrated in our paper, they have numerous important applications including image inpainting, super resolution and deblurring.

Extension to nonlinear measurements is non-trivial for FPS and FPS-SMC. When the measurement operator is non-linear, there is no explicit formulation for $p(x_k \mid y_k, x_{k+1})$, and generating the $\{y_k\}$ sequence without $\{x_k\}$ is challenging. That said, it may be possible to linearize such operators for local approximations to enable FPS and FPS-SMC, which we leave for future work.

Q2: FPS-SMC requires heavy computational cost since it uses multiple particles

A: It is correct that the computational cost for FPS-SMC grows in proportion to particle size. However, modern GPUs excel at parallel computation, and the increase in running time can be significantly smaller than the growth in computational cost. To provide more clarity, we added a paragraph and a table in our revised draft to compare the wall-clock time of FPS and FPS-SMC with multiple diffusion-based solvers for inverse problems (in the appendix). Empirically, we observe that the wall-clock time $t$ of FPS-SMC is approximately $t\propto \sqrt{M}$, where $M$ denotes the number of particles. In practice, we recommend FPS which is twice as fast as DPS, but keeps most of the accuracy of FPS-SMC without requiring more particles.

Q3: Adding more literature review

A: In our initial submission, we placed the related work section in Appendix A due to page limit. In the revised draft, we have included additional discussion on relevant literature in introduction and related work, involving some concurrent work like [1,2]. We have also polished our introduction, background section in the main text.

Q4: Code release

A: We will publish code upon publication.

[1] Gabriel Cardoso, Yazid Janati El Idrissi, Sylvain Le Corff, and Eric Moulines. Monte carlo guided diffusion for bayesian linear inverse problems. arXiv preprint arXiv:2308.07983, 2023.
[2] Luhuan Wu, Brian L Trippe, Christian A Naesseth, David M Blei, and John P Cunningham. Practical and asymptotically exact conditional sampling in diffusion models. arXiv preprint arXiv:2306.17775, 2023.

---

### Meta-Review · Area_Chair_qMgt · 2023-12-05

**Metareview:**

The reviewers agree that there is novelty in the proposed method, even though there are concerns about the incremental improvement over baselines and the lack of comparison with more recent works. The theoretical analysis can also benefit from making the assumptions more realistic.

**Justification For Why Not Higher Score:**

If the proposed method can outperform the SOTA or can be accompanied with a stronger theoretical analysis, then the paper can be stronger.

**Justification For Why Not Lower Score:**

This is really a borderline paper. I tend to suggest its acceptance due to the overall positive feedback, despite some doubts.

---

### Decision · Program_Chairs · 2024-01-16

Accept (poster)